# Inter-Comparison of Field- and Laboratory-Derived Surface Emissivities of Natural and Manmade Materials in Support of Land Surface Temperature (LST) Remote Sensing

**Mary F. Langsdale** [1,*] , **Thomas P. F. Dowling** [1,2] , **Martin Wooster** [1] , **James Johnson** [1] , **Mark J. Grosvenor** [1] , **Mark C. de Jong** [1] , **William R. Johnson** [3] , **Simon J. Hook** [3] **and Gerardo Rivera** [3]

[1] NERC National Centre for Earth Observation (NCEO), c/o Department of Geography, King's College London, London WC2B 4BG, UK; thomas.dowling@unep-wcmc.org (T.P.F.D.); martin.wooster@kcl.ac.uk (M.W.); james.johnson@kcl.ac.uk (J.J.); mark.grosvenor@kcl.ac.uk (M.J.G.); mark.dejong@kcl.ac.uk (M.C.d.J.)

[2] United Nations Environment Programme World Conservation Monitoring Centre, 219 Huntingdon Road, Cambridge CB3 0DL, UK

[3] National Aeronautics and Space Administration-Jet Propulsion Laboratory (NASA-JPL), 4800 Oak Grove Drive, Pasadena, CA 91109, USA; william.r.johnson@jpl.nasa.gov (W.R.J.); simon.j.hook@jpl.nasa.gov (S.J.H.); gerardo.rivera@jpl.nasa.gov (G.R.)

\* Correspondence: mary.langsdale@kcl.ac.uk

**Abstract:** Correct specification of a target's longwave infrared (LWIR) surface emissivity has been identified as one of the greatest sources of uncertainty in the remote sensing of land surface temperature (LST). Field and laboratory emissivity measurements are essential for improving and validating LST retrievals, but there are differing approaches to making such measurements and the conditions that they are made under can affect their performance. To better understand these impacts we made measurements of fourteen manmade and natural samples under different environmental conditions, both in situ and in the laboratory. We used Fourier transform infrared (FTIR) spectrometers to deliver spectral emissivities and an emissivity box to deliver broadband emissivities. Field- and laboratory-measured spectral emissivities were generally within 1–2% in the key 8–12 micron region of the LWIR atmospheric window for most samples, though greater variability was observed for vegetation and inhomogeneous samples. Differences between laboratory and field spectral measurements highlighted the importance of field methods for these samples, with the laboratory setup unable to capture sample structure or inhomogeneity. The emissivity box delivered broadband emissivities with a consistent negative bias compared to the FTIR-based approaches, with differences of up to 5%. The emissivities retrieved using the different approaches result in LST retrieval differences of between 1 and 4 °C, stressing the importance of correct emissivity specification.

**Keywords:** land surface temperature; land surface emissivity; measurement uncertainties; emissivity box method; Fourier transform infrared spectrometer; portable spectrometer

## 1. Introduction

Emissivity is a spectrally varying property of a material, describing at any particular wavelength the efficiency at which an object emits electromagnetic radiation as a function of its temperature. It is mathematically defined as the ratio between the electromagnetic radiation actually emitted by the object at the wavelength in question, and that emitted by a black body at the object's thermodynamic

(or kinetic) temperature [1]. Kirchhoff's law of thermal radiation furthermore states that at any particular wavelength, the absorptivity of a surface is equal to the emissivity of the surface if it is in thermal equilibrium with its surroundings, meaning for example that a perfect blackbody absorbs all the arriving electromagnetic radiation and re-emits the absorbed energy according to Planck's radiation law [2]. However, natural materials are not perfect blackbodies, and most are selective radiators, which may emit electromagnetic radiation according to Planck's radiation law at certain wavelengths, but not others. It is therefore important to understand their spectrally varying emissivity across the electromagnetic spectrum, including within the longwave infrared (LWIR) spectral atmospheric window (8–13 μm) where most remote sensing of land surface temperature (LST) is conducted. This is particularly the case when estimating LST remotely, where knowledge of the target's surface emissivity in the LWIR is essential when converting infrared brightness temperature (BT) measurements into accurate estimates of LST [3].

Emissivity depends on the chemical makeup of a material, and its geometry, surface roughness, and moisture content and as such can show strong seasonally varying cycles and land use/land cover variability [4]. Most soils and vegetation emissivities vary between a minimum of around 0.6 to a maximum of at or close to 1, while pure metals for example can have far lower values [2]. Unfortunately, relatively small errors in the assumed emissivity of a surface can induce quite large impacts on the finally estimated LST. For typical earth surface conditions, Jiménez-Muñoz and Sobrino [5] calculated that emissivity uncertainties of 0.01 typically result in LST uncertainties of around 0.6 K. Given the many applications of LST–such as deriving evapotranspiration and monitoring droughts [6]–recent years have seen an increase in interest in improving the accuracy of LST retrieval as evidenced by the development of new thermal infrared (TIR) sensors capable of LST retrieval such as the ECOsystem Spaceborne Thermal Radiometer Experiment on Space Station (ECOSTRESS) [7] and the classification of LST as an essential climate variable (ECV) by the World Meteorological Organisation's Global Climate Observing System (GCOS) [8]. The correct specification of surface spectral emissivity has been identified as the greatest source of error in current satellite-based measurements of LST [9] and it therefore is essential to try to minimise emissivity uncertainties in order to maximise the accuracy of remotely sensed LST estimates.

Multiple field and laboratory techniques for measuring emissivity have been developed, enabling both spectral emissivity measurement and broadband emissivity retrieval [10–14]. Although unable to perfectly capture field conditions known to impact surface spectral emissivity, such as soil moisture [15] or canopy structure [16], laboratory-based emissivity measurements are often preferred to field-based measurements (for samples that can be transported without modifying the sample and its emissivity). This is because, unlike field measurements, laboratory measurements can be collected under highly controlled conditions, thus reducing errors that might result from changing atmospheric or thermal conditions in the field for example [17]. Online spectral emissivity libraries consist predominately of such laboratory-derived emissivity spectra [18–22]. Data from these spectral libraries have been used extensively to "ground-truth" airborne and satellite LST and emissivity outputs [14,23–25], for the derivation LST algorithm coefficients [26–28] and in the calibration of LWIR satellite and airborne sensors [29]. However, a recent inter-comparison of laboratory emissivity measurements of the same samples reported some quite significant differences in emissivity values from different laboratory measurement setups [30]. For example, they found standard deviations of ±2.52% (0.024) in the emissivities derived for distilled water within the LWIR atmospheric window (8–14 μm). These uncertainties are much larger than those previously reported for laboratory setups [15] and larger than those typically reported with field measurements [11,31,32], thus highlighting the continuing importance of field-based surface emissivity measurements. This is particularly true given that such in situ measurement approaches allow measurements of emissivity under "natural conditions"—for example for samples such as vegetation that is difficult to preserve while transported.

Given that the correct specification of surface spectral emissivity is the greatest source of error in current satellite-based measurements of LST [9], and the discrepancies that have been found both

within laboratory and between field and laboratory measurements detailed above, there is a need for further rigorous examination of the degree of agreement between current approaches to emissivity measurement. With this in mind, we conducted a study to compare different field and laboratory spectral emissivity measurement approaches, using the same targets to better understand the emissivity differences that can result from use of different measurement approaches and/or different measurement conditions. We have focused on Fourier transform infrared (FTIR) spectrometer-based emissivity measurement systems since these are the most common type used to provide spectral emissivity measurements, applying to the measured spectra a variety of different post-processing approaches to derive the surface emissivity information. We also include a comparison of these spectrally resolved data to the broadband emissivities produced using an "emissivity box", a popular low-cost method of broadband field emissivity determination that uses a sequence of LWIR radiometer measurements and a specially constructed box [33]. The impact of the emissivity measurement uncertainties from these methods on calculation of in situ LSTs is assessed as the last stage of our investigation.

## 2. Emissivity Measurement Techniques

Summarised in Table 1 are different field emissivity measurement techniques deployed in previous studies. The most utilised are variants of the emissivity box method, detailed in Rubio et al. [33,34], which provide broadband LWIR emissivity estimates, and approaches based on spectral radiance measurements made by field portable FTIR spectrometers, which provide spectrally resolved LWIR emissivity data [31]. As detailed by Rubio et al. [33,34], the two primary variants of the box method are the two-lid approach [35] and the one lid approach [36]. Both involve a bottomless box with highly reflective (for example polished aluminium) inner walls and a LWIR radiometer to make the broadband measurements. During each measurement, the box is covered by a lid with a small central hole through which the radiometric measurements are made, with the lid having either high reflectance (the "cold lid") or high emissivity (the "hot lid"). A sequence of four radiometer measurements with the box and lids in different configurations provide the data to estimate the broadband emissivity of the surface over which the box is placed [33,34].

**Table 1.** Overview of various different field emissivity measurement techniques, with variants of the first approaches considered in this study.

| Method | Overview | References |
|---|---|---|
| Emissivity Box Method | One- and two-lid variants of the emissivity box method, used to determine the LWIR broadband emissivity of a surface | [33–39] |
| Portable FTIR Spectrometer Approach | Use of field-portable Fourier Transform Infrared (FTIR) spectrometer to estimate LWIR surface spectral emissivity | [31,32,40,41] |
| Temperature and Emissivity (TES) retrieval algorithm applied to a multi-band radiometer | Application of the Advanced Spaceborne Thermal Emission and Reflection Radiometer (ASTER) Temperature and Emissivity Separation (TES) algorithm with in situ radiance measurements obtained using multi-band radiometers | [42,43] |
| Novel Emissiometer | Novel instrument combining an oscillating TIR radiance source with digital signal processing to determine the band-effective emissivity of a radiometer | [12,44] |
| Sun Shadow Method | Similar approach to day/night LST retrieval algorithm adapted to in situ measurements in sun and sun-shadow with spectroradiometer to derive spectral emissivities | [45,46] |

The field spectrometer approach to the emissivity measurement is detailed by Salvaggio and Miller [32], and involves the spectrometer measuring the emitted LWIR signal from the surface and using this, along with a measurement of the downwelling LWIR atmospheric radiation, to derive the surface's spectral emissivity. The downwelling component is most commonly assessed using a downward looking measurement of a gold Lambertian panel, which reflects almost all of the LWIR atmospheric radiation irradiating it.

In addition to field emissivity approaches, there exist a number of laboratory-based methods to assess surface emissivity, generally based on FTIR spectroscopic techniques, which provide surface spectral emissivity values. The spectrometers measure either LWIR sample emission or directional hemispherical reflectance (DHR) [10]. In the emission mode the emissivity estimate is derived through comparison of the spectral radiance emitted by the sample to that emitted by a blackbody at the same temperature (for example [19]). Being lab-based, this approach generally means the sample must be heated to temperatures significantly above the laboratory such that any low emissivity features in the resulting emissivity spectra are not simply "filled in" by reflected LWIR radiation coming from the surroundings at the same temperature as the sample. Consequently, the method is unsuited to samples such as vegetation [47]. To avoid this, FTIR spectrometers operating in the DHR mode are used, generally with a source of intense LWIR radiation that is used to illuminate the sample and assess its LWIR reflectance via consecutive measurements of the sample and a highly reflective reference standard such as Infragold [48]. Emissivity is then calculated from the LWIR reflectance spectra using Kirchhoff's law [49].

Many field emissivity and LST validation studies have used the box method since the equipment is relatively simple, inexpensive, easily portable, and with minimal power requirements (e.g., [11,14,50–54]). Multiple studies have assessed the quality of emissivities derived using the approach, typically by comparing them to full spectral emissivity data coming from laboratory measurements convolved to the waveband of the LWIR radiometer used in the box [15]. The conclusions of these studies generally indicate that the quality of the box-derived field emissivity data is highly dependent on the measurement conditions, particularly for the one-lid variant [11,15,33]. Under favourable measurement conditions, a strong degree of agreement is seen between the data derived by the box method and that of the various laboratory spectral measurement approaches applied. Mira et al. [15] and Nerry et al. [38] for example both observed that the two-lid box method produced broadband LWIR emissivity estimates with a mean error of ±0.5% under stable field conditions (low winds and constant cloud conditions that help keep the sample surface temperature consistent while the measurements are made). Göttsche and Hulley [11] reported less than 1% difference for sand samples in Gobabeb (Namibia) where clear, cold skies with low winds made measurement conditions optimum. However, under less suitable conditions (e.g., high winds and variable cloud cover), the sensitivity of the derived surface emissivity value to changes in the sample temperature during the measurement can result in large errors. A change of 3 K over the measurement period results in emissivity errors of up to 2% in the one-lid method for example [34]. While such percentage errors seem small, due to coupling of LST and emissivity, a 1% error in specified emissivity will generally result in about a 0.5 K error in the derived LST [9]. Hence the accuracy of surface emissivity data is key to accurate LST derivation.

Compared to the box method, fewer studies exist comparing field- and lab-derived emissivity data based on FTIR spectrometer measurements [31,32,47,55,56]. However, as with the box method, the studies that have been conducted found that the accuracy of the field-derived data is highly dependent on the environmental conditions that existed during the measurement. For example, Salvaggio and Miller [32] assessed the field spectral emissivity data coming from measurements made with the Designs and Prototypes (D&P) μFTIR system, specifically designed for field emissivity measurement. Under ideal measurement conditions (stable, low winds and clear skies), the mean absolute emissivity error was less than 1% for most surface samples, with the D&P spectral measurements processed to spectral emissivity using Horton et al.'s [57] spectral smoothness approach. However, more problems were observed with measurements made under conditions of high humidity

and air temperature, and/or more variable conditions [31,55]. Horton et al. [57] found that a 0.5 K change in sample temperature during the measurement procedure resulted in errors in the final calculated emissivity of 2.5%. As a result, samples with relatively low thermal inertia (such as dry soils) or samples that experience rapid evaporative cooling in the near-surface layer (such as water, damp soils, or dewy vegetation) can show higher errors under changing environmental conditions, such as high winds [17].

As well as these meteorological factors, observed differences between laboratory and field emissivity measurements (whether the latter be from the box- or FTIR-based approaches) are often attributed to physical changes in the sample, which may occur between the field and the laboratory, for example in terms of its structure and surface moisture [15]. Such possibilities for error further highlight the importance of field emissivity measurements. However, since the accuracy assessment of the field methodologies is often performed through comparison with laboratory-derived measurements, any differences between the laboratory and field sample conditions can affect the evaluation. Studies that intercompare the emissivities of the same samples derived by different field measurement approaches may help to redress this issue, but few such studies exist. Those that have been conducted considered are restricted to few sample types (e.g., soils or sands) or have been based on rather limited comparisons, for example due to differing instrument spectral responses [12]. A critical finding is that of Mira et al. [15], who observed emissivity differences between 2% and 7% in the 8–9 μm LWIR band between the values derived using the two-lid emissivity box and the TES-retrieved radiometer approach (see Table 1), corresponding to a 0.7–2.6 K error in derived LST.

## 3. Methods

Measurements were made of multiple manmade and natural samples with varying physical structures during two field campaigns in the UK and Italy using four methods: (i) a laboratory FTIR spectrometer setup at King's College London operating in DHR mode, (ii–iii) two portable field FTIR spectrometers with different processing approaches, and (iv) a two-lid emissivity box constructed at King's College London.

### 3.1. Instrumentation, Measurements, and Post-Processing

#### 3.1.1. Emissivity Determination Using the Laboratory FTIR Spectrometer

In the laboratory, high spectral resolution (4 cm$^{-1}$) surface emissivity spectra of the target samples were derived from directional hemispherical reflectance LWIR spectral measurements made by a Bruker Vertex 70 FTIR spectrometer with an external highly reflective gold integrating sphere and an external thermal infrared source, as shown in Figure 1. The full measurement setup is detailed in Langsdale et al. [30], and the measurements covered the spectral range 2.5–16 μm, extending beyond the normal LWIR atmospheric window (8–13 μm).

The data coming from the laboratory system shown in Figure 1 can be processed to surface spectral emissivity using either the substitution or comparative methods, which are detailed in Hecker at al. [58]. The authors of [30] found that surface spectral emissivities derived using the comparative method, which uses the internal wall of the diffusely coated gold integrating sphere as the reference target, on the same laboratory system were within 1.5% of the mean of those derived by a wide range of international laboratories' measurements (spectral range 8–14 μm). To measure emissivity using this comparative method, the target surface was placed directly underneath the sample port of the external integrating sphere and illuminated with the LWIR beam coming from the external source. The reflected spectral radiance ($V_s(\lambda)$) was then measured and compared to a subsequent measurement of the reflected radiance from the internal wall of the integrating sphere ($V_r(\lambda)$), enabling the calculation of sample reflectance ($\rho_s(\lambda)$):

$$\rho_s(\lambda) = \frac{V_s(\lambda) - V_o(\lambda)}{V_r(\lambda) - V_o(\lambda)}\rho_r(\lambda) \tag{1}$$

where $V_o(\lambda)$ is an open port measurement as detailed in Hecker et al. [58] and $\rho_r(\lambda)$ is the absolute reflectance of the internal gold wall of the integrating sphere ($\rho_r(\lambda) \approx 0.97$ across 2.5–14 μm) used as the reference target. An internal rotating mirror was used to move the infrared beam illumination between the sample and the reference position. Sample spectral emissivity ($\varepsilon_s(\lambda)$) was then calculated from reflectance using Kirchhoff's law [59]:

$$\varepsilon_s(\lambda) = 1 - \rho_s(\lambda) \tag{2}$$

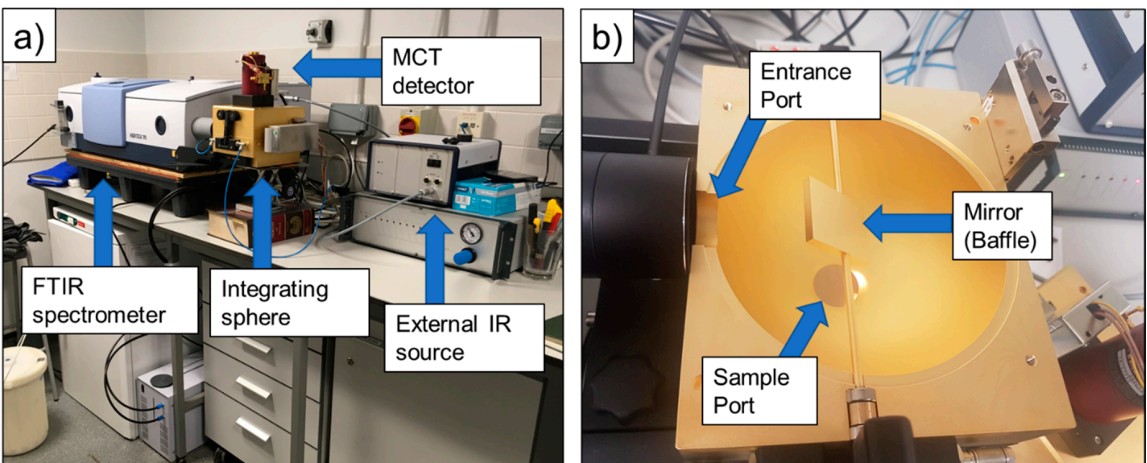

**Figure 1.** (**a**) Laboratory setup for surface spectral emissivity determination based on measurements made by a Bruker Vertex 70 FTIR spectrometer installed at King's College London along with an external water-cooled longwave infrared (LWIR) radiation source and a gold-coated integrating sphere. (**b**) Details of the inside of the integrating sphere, showing the gold coating used as the reference target in the comparative method and rotating mirror to direct the measurement beam from entrance port to sample port/internal wall.

For each sample, a minimum of the three emissivity measurements was collected to enable consideration of measurement variability. More measurements were made for low reflectance samples (card, grass and water) and for inhomogeneous samples (gravel and grass). Each individual spectral measurement consisted of either 500 or 1000 coadded scans, with the higher number of scans used for samples with low LWIR reflectances.

### 3.1.2. Emissivity Determination Using the Field Portable FTIR Spectrometers

Two portable FTIR spectrometers were deployed in this study (Table 2). The first was the aforementioned D&P μFTIR spectrometer [31,32,40], specifically designed for surface emissivity measurement in the field (Figure 2). The instrument operates in passive emission mode to measure emitted LWIR radiation, and uses the two-temperature blackbody approach for its calibration. It has a 45° mirror (rotating to allow angled measurements) within an enclosed tube. The main improvement on the μFTIR design described originally in Korb et al. [31] and Hook and Kahle [40] is the inclusion of a Stirling cycle cooling for the detector in place of liquid nitrogen. The second FTIR deployed was a Bruker EM27, also with a Stirling cycle cooled detector and an internal blackbody target that can be rapidly heated and cooled to provide the necessary two point calibration [30]. Though this system is designed primarily for atmospheric remote sensing, it is easily adapted to assess surface emitted LWIR radiation via attachment of a 45° flat high IR reflectance gold mirror as shown in Figure 2. This mirror can then be used to reflect the surface target emitted and gold-panel reflected LWIR radiation into the spectrometer. The system, its 12 V battery/inverter and a controlling laptop were mounted on a rugged trolley for relatively easy transport around a field site. Spectral resolutions used were the maximum

for the two systems, namely 0.5 cm$^{-1}$ for the EM27 and 4 cm$^{-1}$ for the D&P µFTIR, with spectral sampling intervals in practice of 0.25 cm$^{-1}$ and 3 cm$^{-1}$. The D&P was available for the measurements in Italy only.

**Table 2.** Instrument specifications for the portable FTIR spectrometers deployed herein, namely a Bruker EM27 FTIR spectrometer and the Designs and Prototypes µFTIR [31,40,60].

| Parameter | Bruker EM27 | Designs and Prototypes µFTIR |
|---|---|---|
| Spectral Range | 4.5–14.3 µm (700–2200 cm$^{-1}$) | 2.0–14.0 µm |
| Spectral Resolution | 0.5 cm$^{-1}$ | 4 cm$^{-1}$ |
| Sampling Rate | 0.25 cm$^{-1}$ | 3 cm$^{-1}$ |
| Interferometer | Michelson | Michelson |
| Detector | HgCdTe | HgCdTe |
| Dimensions (cm) | 40 × 36 × 27 | 33 × 46 × 32 |
| Weight | 18 kg | 12.4 kg |
| Power | 40 W (Average), 80 W (Max) | 18 W |
| FOV (@ 1 m) | 1.7° (60 mm) | 4.8° (78 mm) |

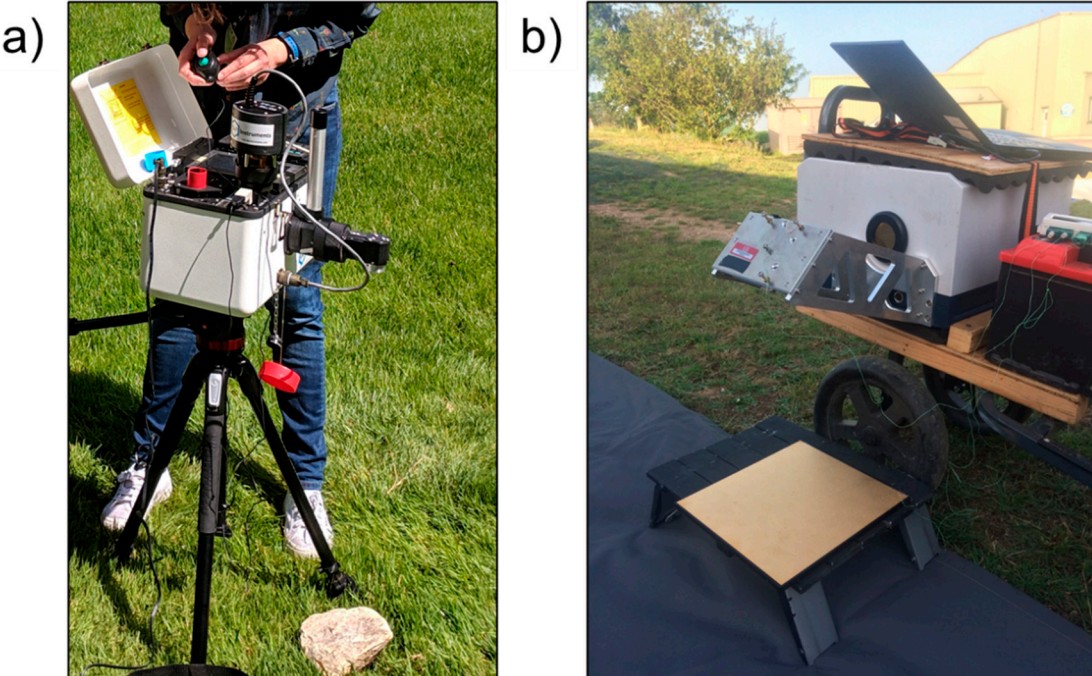

**Figure 2.** (**a**) Field-deployed Designs and Prototypes µFTIR spectrometer and (**b**) field-deployed Bruker EM27 FTIR spectrometer with a 45° mirror attachment fitted to view upwelling radiation from the surface, here shown assessing downwelling LWIR atmospheric radiation via observations of an Infragold panel.

To retrieve surface spectral emissivities from the passive LWIR spectra collected by either of the FTIR instruments, the calibration blackbody temperatures were first set to appropriately bracket the sample temperature [17]. As recommended by Salvaggio and Miller [41], the hot and cold blackbody temperatures used for the calibrations were set to approximately 10 K above and below the estimated sample temperature to reduce extrapolation error, although very high ambient air temperatures encountered at the Italian field site required the cool blackbody to be elevated above this limit. Sample temperature was itself estimated using a FLIR i7 handheld LWIR thermal imaging camera. Consecutive spectral measurements were then made of the sample (*L*) and of a 13 cm × 13 cm Labsphere Infragold panel (*L*$_{panel}$), with the panel measurement used as a proxy for the downwelling

LWIR atmospheric signal. The panel has a known and spectrally flat emissivity ($\varepsilon_{panel}$), provided by the manufacturer as $0.03 \pm 0.01$ across 2.5–14 µm range. The panel was placed in the same configuration as the sample, positioned just above the sample location.

The D&P µFTIR spectrometer comes with its own software to estimate sample emissivity across 7.5–12.0 µm from these measurements. Sample temperatures are estimated using the "Maximum Spectral Temperature" method detailed in Salvaggio and Miller [32] and developed by Korb et al. [31] and Hook and Kahle [40]. Emissivity uncertainties were taken as the standard deviation of multiple measurements. The measurement procedure takes around 25 min for a complete set of measurements. For the EM27 we developed our own emissivity measurement approach and software, with the full measurement sequence (three consecutive and repeated measurements of the sample and Infragold panel along with spectral calibration) typically taking around 20 min. From the measurements of the gold panel, downwelling radiance ($L^{\downarrow}$) spectra were first estimated as:

$$L^{\downarrow}(\lambda) = \frac{L_{panel}(\lambda) - \varepsilon_{panel}(\lambda)L_{BB}\big(T_{panel}, \lambda\big)}{1 - \varepsilon_{panel}(\lambda)} \tag{3}$$

where $T_{panel}$ is the kinetic temperature of the Infragold panel (K), measured with a contact k-type thermocouple (manufacturer-stated accuracy ±0.1 K) and $L_{BB}\big(T_{panel}, \lambda\big)$ is the blackbody spectral radiance at temperature $T_{panel}$ calculated using the Planck function such that:

$$L_{BB}\big(T_{panel}, \lambda\big) = \frac{2hc^2}{\lambda^5\left(e^{\frac{hc}{\lambda k T_{panel}}} - 1\right)} \tag{4}$$

where $h$ is the Planck constant ($6.62606957 \times 10^{-34}$ Js), $c$ is the speed of light ($299,792,458$ ms$^{-1}$), and k is the Boltzmann constant ($1.3806488 \times 10^{-23}$ $JK^{-1}$).

If the sample temperature ($T_s$) is accurately known, the surface spectral emissivity ($\varepsilon(\lambda)$) of the sample can be retrieved through use of a rearranged radiative transfer equation appropriate to a surface-viewing sensor positioned close to the target [32]:

$$\varepsilon(\lambda) = \frac{L(\lambda) - L^{\downarrow}(\lambda)}{L_{BB}(T_s, \lambda) - L^{\downarrow}(\lambda)} \tag{5}$$

where $L_{BB}(T_s)$ is the blackbody spectral radiance at temperature ($T_s$). However, sample temperature can vary even over short timescales (e.g., due to wind), and can be hard to assess accurately in the field for certain targets (e.g., vegetation) even under good measurement conditions. We therefore avoided having to specify $T_s$ by using the "spectral smoothness" approach [57], an approach determined as optimal for emissivity derivation based on field-portable FTIR measurements by Salvaggio and Miller [32]. We implemented this by identifying a realistic sample temperature range as in Salvaggio and Miller [32] and calculating emissivity using Equation (5) for all temperatures within this range (in increments of 0.01 K). The sample temperature was taken to be the temperature, which minimised residuals in the resulting emissivity spectra that were clearly associated with atmospheric absorption and emission features in the 8.12–8.60 µm range of the short wavelength lobe of the silicate doublet. Emissivity uncertainties for the EM27 were calculated through propagation of the uncertainties in the input parameters. Uncertainties in the gold panel temperature and emissivity were taken as the manufacturer-stated accuracy (0.01 K and 0.01 respectively), which in the sample temperature as the 0.01 K precision, and that in the sample and gold panel spectra as the standard deviation of the measurement coadds (minimum 12 per spectra). Due to its extremely low emissivity, no adjustment was made for self-emission of the 45° mirror used to direct LWIR radiation from the sample to the spectrometer, nor for errors related to the fact that the cold sky temperatures are far lower than the minimum temperature of the two-point blackbody calibration. The typically low values of the

downwelling radiation from the cold (clear) sky compared to the LWIR emission from the samples, along with the high emissivities (and thus low reflectances) of the samples, mean that uncertainties in the assessment of the downwelling sky radiation do not have much impact on the final emissivity uncertainty [31].

### 3.1.3. Emissivity Determination Using the Emissivity Box

We constructed a two-lid emissivity box at King's College London (Figure 3), based on the design of Rubio et al. [33,34] with (i) an improved outer thermal insulation layer surrounded by a robust outer case, (ii) a 3D-printed angled port to hold the radiometer at a constant view zenith angle of 5° to reduce the "Narcissus" effect, as in Göttsche and Olesen [37], (iii) continuous 1 Hz sampling of the radiometer measurements as in Göttsche and Olesen [37] to enable identification and rejection of erroneous readings (e.g., when conditions were not stable) during post-processing readings, and (iv) the addition of a "heating tray" to help the hot lid more quickly achieve its optimal temperature, while also reducing heat loss in cooler conditions. While our design could also be used for the one-lid approach, the two-lid method is considered more robust in windy or otherwise variable field conditions [34], some of which we encountered during our study.

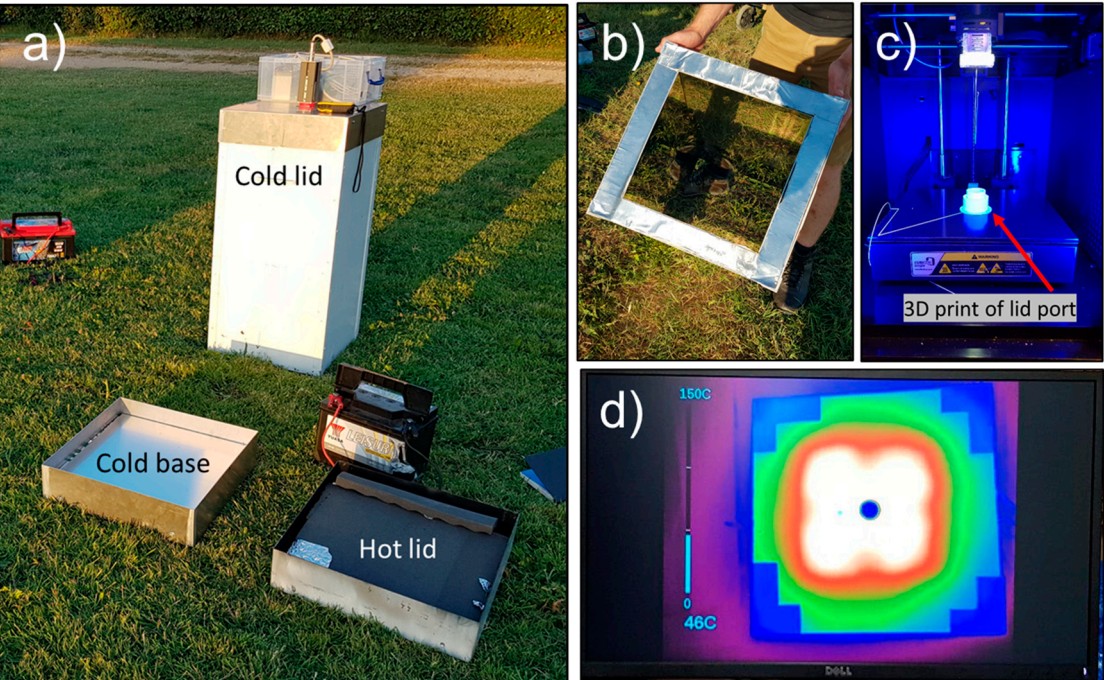

**Figure 3.** (**a**) Insulated emissivity box with the cold base and battery-powered hot lid. Details include (**b**) internal walls of highly polished aluminium, (**c**) a 3D printed 5° radiometer port for consistent off-centre angled sampling to avoid the "Narcissus" effect, and (**d**) evenly distributed electronic heating pads on the hot lid enabling heating up to at least 60 °C when combined with the heating tray.

Broadband surface emissivity was determined using our emissivity box via a sequence of BT measurements made with a Heitronics KT15.85 IIP radiometer fitted in the angled radiometer port to sequentially view the target surface and the base, as described in [31]. This radiometer is the same model as that used at the four permanent LST validation stations described in Göttsche et al. [50] and operates over the spectral range 9.6–11.5 µm, which is located well within the LWIR atmospheric window (Figure 4). BT measurements from the radiometer ($T$, kelvin) were converted into spectral radiances ($L$, $Wm^{-2}sr^{-1}µm^{-1}$) using Planck's radiation law (Equation (4)) evaluated at the effective radiometer central wavelength (approximately 10.55 µm; the exact value depending on the target temperature). Laboratory calibration tests confirmed the radiometer to have an absolute accuracy of

±0.5 K plus 0.7% of the difference between the BT of the target and the radiometer body temperature (taken as the ambient temperature). For example, if the ambient temperature was 300 K and the target BT was recorded as 295 K, the absolute accuracy was determined to be ±(0.5 + [0.7/100 × 5]) = ±0.535 K. When fitted into the angled port, the observed surface area was 170 cm$^2$.

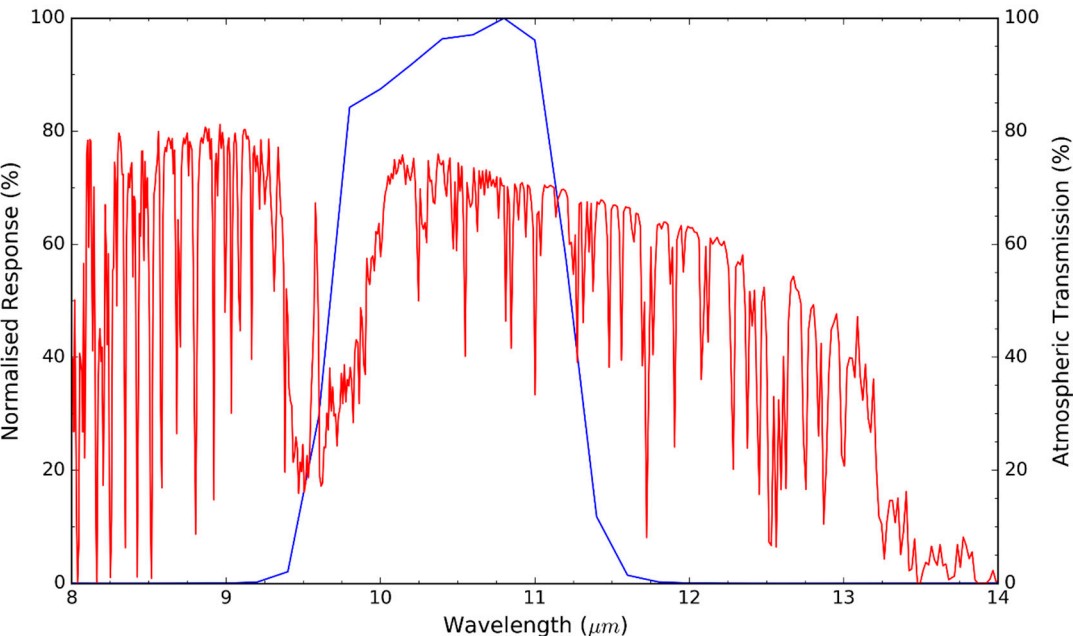

**Figure 4.** Spectral response function of the Heitronics KT15.85 radiometer (blue, left axis) overplotted on the atmospheric transmission of a standard mid-latitude summer atmosphere (red, right axis) calculated using MODTRAN 5.0 [61].

To make the measurements necessary to estimate the targets surface emissivity, the box was first placed on the target surface sample and left for two minutes to ensure stabilised temperatures. Measurements then proceed as in Figure 5, with the rationale for this sequence explained in Rubio et al. [33]. Using the same nomenclature as in Rubio et al. [33] and Figure 5 the broadband emissivity ($\varepsilon_0$) of the target surface if the box were ideal was calculated as:

$$\varepsilon_0 = \frac{L^3 - L^1}{L^3 - L^2} \tag{6}$$

where (in order of measurement), $L^2$ is the sample radiance measured when the box is over the ground sample with the cold lid in use, $L^1$ the sample radiance made with the hot lid in use instead of the cold lid and $L^3$ the radiance obtained when putting the box with the hot lid on over a cold base with the same emissivity as the cold lid.

However, the box departs from non-ideal behaviour (because the emissivity of the hot lid cannot be 1 and the emissivity of the cold lid cannot be 0) as detailed by Rubio et al. [34], who developed a correction factor ($\delta\varepsilon$) for these effects equal to:

$$\delta\varepsilon = (1 - \varepsilon_0)\left(1 - \frac{\left(L^3 - L^2\right)(1 - \varepsilon_c)}{(L^3 - L^2) - (L^3 - L^1)P + (L^2 - B_c)Q}\right) \tag{7}$$

where $\varepsilon_c$ is the emissivity of the polished aluminium, the term $B_c$ the radiance measured through the box with the cold lid on top and cold base below (effectively equal to the blackbody spectral radiance at the temperature of the aluminium), and $P$ and $Q$ are box-specific parameters between 0 and 1 defined

by the box geometry and material properties such that $P = f(\varepsilon_c, \varepsilon_h)$ and $Q = g(\varepsilon_c)$ (see [34] for exact expressions of $P$ and $Q$).

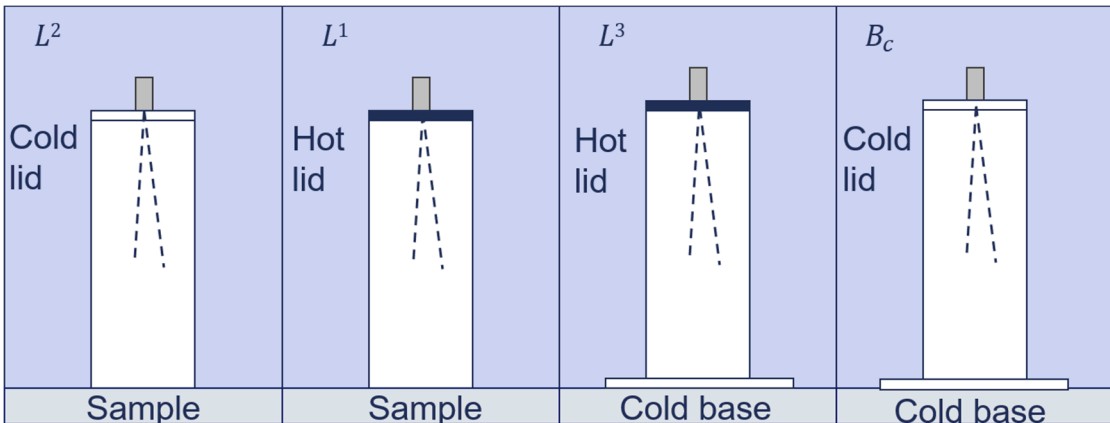

**Figure 5.** Radiance measurement sequence for the two-lid variant of the emissivity box method, using nomenclature from Rubio et al. [33]. Measurements proceed from left ($L^2$) to right ($B_c$).

The final broadband emissivity estimate of the target sample surface is then given by the sum of the outputs from Equations (6) and (7):

$$\varepsilon = \varepsilon_0 + \delta\varepsilon \tag{8}$$

The emissivity of the hot lid is that of the high emissivity paint it was covered in (provided by the manufacturer as $\varepsilon_h = 0.98$). Prior to field deployment, measurements were conducted to determine the emissivity of the cold lid as derived in Appendix 1 of Rubio et al. [34]:

$$\varepsilon = \frac{B(T_{rad})}{\sigma T_{kin}^4} \tag{9}$$

where $B(T)$ is the spectral radiance derived from Planck's radiation law at temperature $T$ (kelvin) as in Equation (4) and $\sigma$ is the Stefan–Boltzmann constant $\left(5.67 \times 10^{-8} \mathrm{Js^{-1}m^{-2}K^{-4}}\right)$. Through this method, the emissivity of the polished aluminium ($\varepsilon_c$) of the cold lid was determined as 0.05, resulting in P and Q as 0.0123 and 0.4223 respectively for the box deployed herein. These values were derived again based on measurements made at the end of each field campaign to identify any changes, associated for example with oxidisation of the box interior aluminium surface or damage to the paint of the hot lid, but no such changes were found.

A minimum of five repeated measurements were collected per sample, with averages and standard deviations of the multiple measurements calculated. Uncertainties for each sample were taken as the standard deviation of the multiple measurements. Example values for the two-lid box method developed at King's College London are shown in Table 3.

**Table 3.** Example values for the two-lid box measurements, with gravel, grass, and sand shown. Measured temperatures are expressed in kelvin and equivalent radiances in brackets in $\mathrm{Wm^{-2}sr^{-1}\mu m^{-1}}$.

| Sample | $L^2$ | $L^1$ | $L^3$ | $B_c$ | $\varepsilon_0$ | $\delta\varepsilon$ | $\varepsilon$ |
|---|---|---|---|---|---|---|---|
| Gravel | 303.25 (10.27) | 304.02 (10.39) | 318.36 (12.77) | 304.59 (10.48) | 0.952 | 0.001 | 0.953 |
| Grass | 299.23 (9.65) | 299.81 (9.74) | 317.98 (12.71) | 304.02 (10.39) | 0.971 | −0.002 | 0.969 |
| Sand | 304.02 (10.39) | 305.74 (10.66) | 322.38 (13.49) | 303.44 (10.30) | 0.913 | 0.004 | 0.917 |

### 3.2. Sites and Experimental Samples

Field measurements for the study were made in three locations: a disused airfield at Alconbury (UK), a farm in Grosseto (Italy), and Duxford Aerodrome (UK; Figure 6). In total, fourteen different surface samples were considered, detailed in Table 4 and shown in Figure 7. These included manmade samples such as a large tarpaulin used as calibration targets in an accompanying airborne campaign, and natural samples such as homogeneous areas of grass, sand, and water.

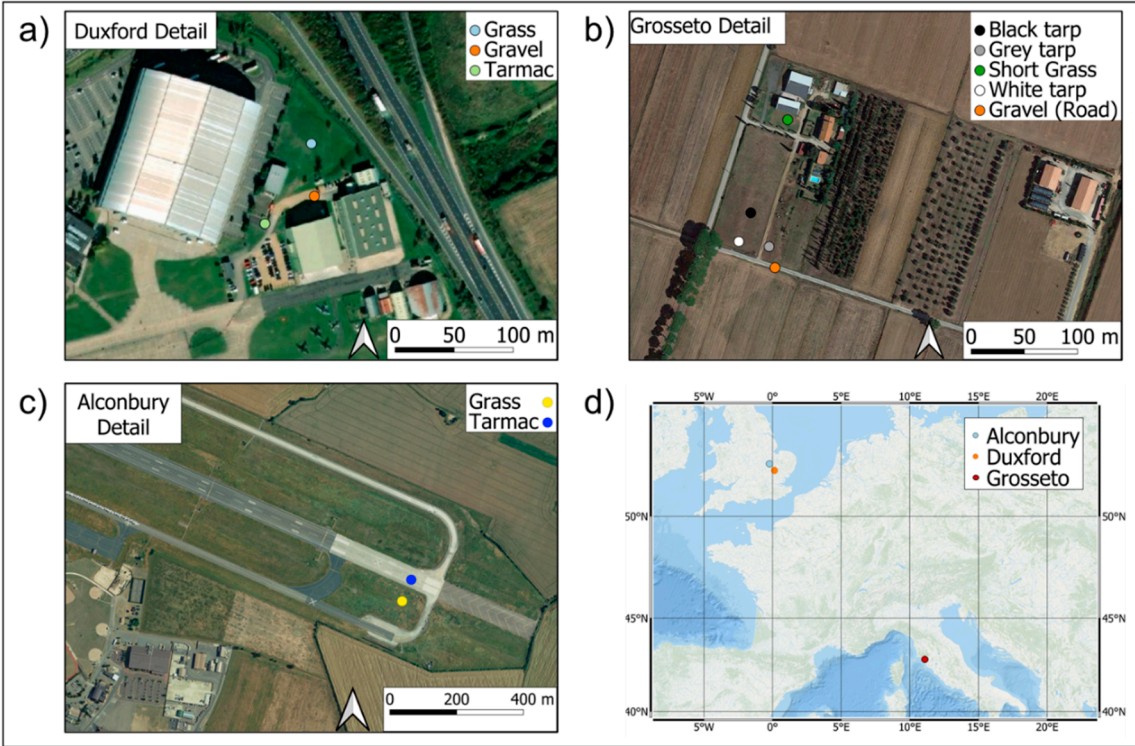

**Figure 6.** Sites in the UK and Italy where the study measurements took place, showing (**a**) detail of Duxford where measurements of grass, gravel, tarmac, and the tarpaulins were collected; (**b**) detail of Grosseto indicating locations of the same tarpaulin measurements there; (**c**) detail of the runway area at Alconbury, showing the grass and the tarmac where all non-grass samples were placed for field measurement, and (**d**) the relative locations of the three field sites.

Field measurements were collected in the late afternoon at all sites to try to maximise thermal contrast between target radiance and downwelling radiance, as recommended in Salvaggio and Miller [41]. All data were collected under stable and generally low wind, clear sky conditions, with these considered suitable for application of the two-lid box method. Air temperatures at the UK sites were between 19 and 25 °C, with wind speeds recorded in Alconbury and Duxford of $3.6 \pm 0.9$ ms$^{-1}$ and $4.3 \pm 0.3$ ms$^{-1}$ respectively. Air temperatures in Italy were higher (between 30 and 32 °C) with a recorded wind speed of $2.3 \pm 0.5$ ms$^{-1}$. Relative humidity throughout the measurement period was 44% ± 4%, 56% ± 6%, and 60% ± 7% in Alconbury, Duxford and Grosseto respectively.

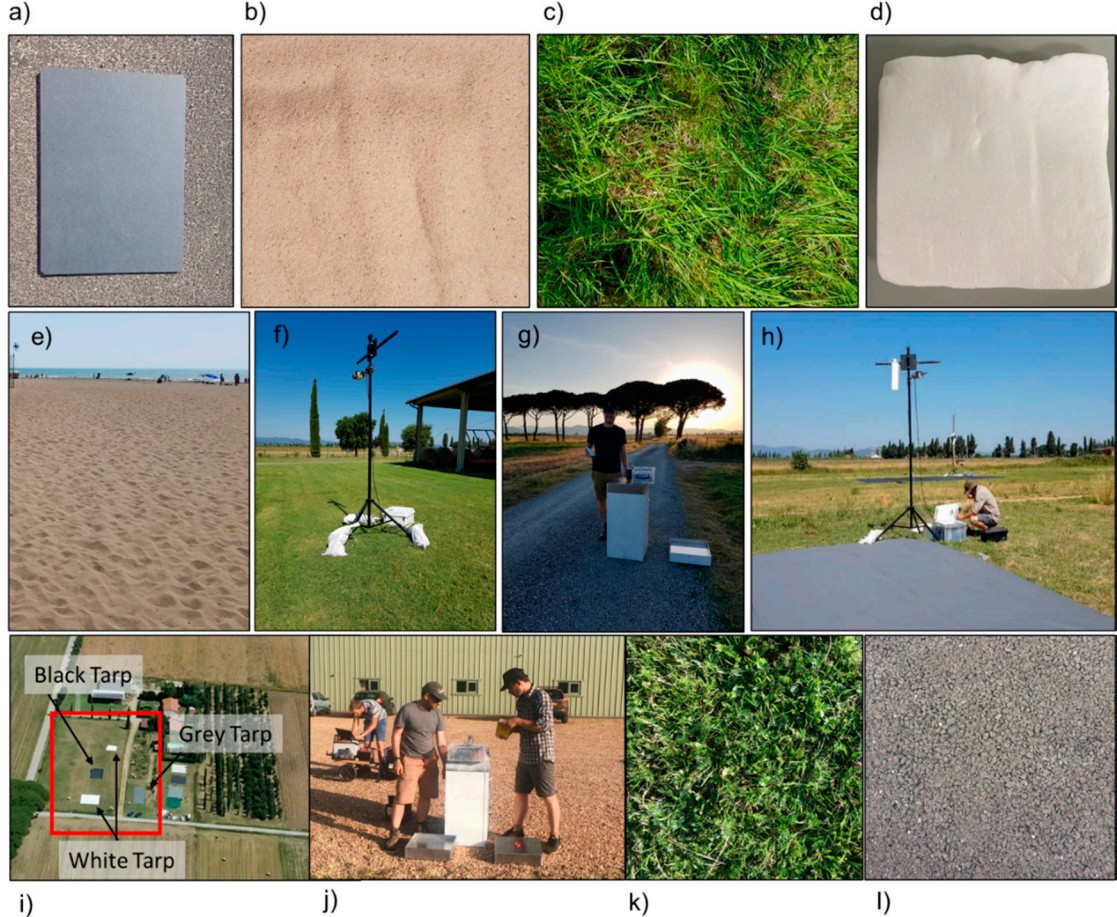

**Figure 7.** All samples considered in this study other than distilled water. Samples (**a**) to (**d**) were measured for the Alconbury comparison and show (**a**) black hardboard card on the tarmac, (**b**) construction sand, (**c**) green grass in Alconbury, and (**d**) polystyrene. Samples (**e**) to (**l**) were measured in Italy, with (**e**) the beach near Grosseto from where sand was collected, (**f**) the short green grass in Grosseto, (**g**) the gravel drive in Grosseto with emissivity box pictured, (**h**) a close-up the grey tarpaulin in Grosseto, (**i**) the white, grey and black tarpaulin photographed from the plane in Grosseto, Italy while in use as calibration targets, (**j**) gravel in Duxford during measurement, (**k**) green grass in Duxford, and (**l**) tarmac in Duxford.

**Table 4.** Samples considered for the field and laboratory emissivity inter-comparison, with the number of measurements made using each indicated in brackets in the final column. Note that the three tarpaulins were measured by the EM27 in both Duxford (UK) and Grosseto (Italy).

| Sample Description | Sample ID | Location (Field) | Date | Instruments | |
|---|---|---|---|---|---|
| | | | | **Field** | **Lab** |
| Black hardboard card (5 mm × 240 mm × 303 mm) | Card | Alconbury, UK | May-18 | EM27 (3) | Vertex (5) |
| Polystyrene (40 mm × 160 mm × 150 mm) | Polystyrene | Alconbury, UK | May-18 | EM27 (3) | Vertex (4) |
| Green grass (max. 200 mm height) | Grass_Alc | Alconbury, UK | May-18 | EM27 (3) | Vertex (5) |
| Construction sand | Sand_Alc | Alconbury, UK | May-18 | EM27 (3) | Vertex (3) |
| Distilled Water | DistilledWater | Alconbury, UK | May-18 | EM27 (3) | Vertex (4) |
| Sandy gravel drive | Gravel_Gro | Grosseto, Italy | Jun-19 | Box (5), EM27 (1), D&P (3) | - |
| Short dry grass | Grass_Gro | Grosseto, Italy | Jun-19 | Box (5), EM27 (1), D&P (3) | - |

**Table 4.** *Cont.*

| Sample Description | Sample ID | Location (Field) | Date | Instruments | |
|---|---|---|---|---|---|
| | | | | Field | Lab |
| Beach sand | Sand_Gro | Grosseto, Italy | Jun-19 | Box (5), EM27 (1–Grosseto, 3–Duxford) | Vertex (3) |
| White PVC tarpaulin (630 gsm) | WhiteTarp | Duxford, UK (Box, EM27); Grosseto, Italy (EM27) | Jun-19 | Box (5), EM27 (1–Grosseto, 3–Duxford) | Vertex (3) |
| Grey polyester tarpaulin (matte finish) | GreyTarp | Duxford, UK (Box, EM27); Grosseto, Italy (EM27; D&P) | Jun-19 | Box (5), EM27 (1–Grosseto, 3–Duxford), D&P (3) | Vertex (3) |
| Black polyester tarpaulin (matte finish) | BlackTarp | Duxford, UK (Box, EM27); Grosseto, Italy (EM27) | Jun-19 | Box (5), EM27 (1–Grosseto, 3–Duxford) | Vertex (3) |
| Gravel driveway (gravel pieces 10–40 mm) | Gravel_Dux | Duxford, UK | Jun-19 | Box (5), EM27 (3) | Vertex (6) |
| Short green grass mixed with clover | Grass_Dux | Duxford, UK | Jun-19 | Box (5), EM27 (3) | - |
| Homogeneous road tarmac | Tarmac | Duxford, UK | Jun-19 | Box (5), EM27 (3) | - |

### 3.2.1. Sample Preparation

For the EM27-based field emissivity measurements conducted in Alconbury in May 2018, the black card, construction sand, and polystyrene (Figure 7a,b,d) were placed onto the runway tarmac shown in Figure 6c for measurement. For measurement of the sand, an area greater than the instrument FOV and with a depth of at least 3 cm was prepared on the tarmac. Distilled water was poured into a plastic tray to a depth of 15 mm for measurement, while the grass measurement (Figure 7c) was conducted on the vegetated area neighbouring the runway as indicated in Figure 6c.

For the other field-based emissivity measurements, all samples aside from the beach sand (Sand_Gro) were measured as found and as shown in Figures 6 and 7. A sample of beach sand (Sand_Gro) was collected from the beach shown in Figure 7e for measurement the following day at the same time as the other targets. The emissivity measurements of the tarpaulins, which were being used as calibration targets for airborne remote sensing measurements, were collected when the tarpaulins were laid out for the overhead flights as shown in Figure 6b.

For the laboratory emissivity measurements, flat samples such as the tarps, card, and polystyrene were placed under the sample port of the integrating sphere shown in Figure 1a, with no gap between port and sample. To preserve the structure and moisture content of the Alconbury grass sample, a section of turf was extracted (Figure 8a–b) and measured the following morning in a foil container of high reflectivity, with the grass pressed underneath the sample port while ensuring no blades went inside the integrating sphere. This method was chosen over the method detailed in Salisbury and D'Aria [48] as it better mimicked field conditions. Distilled water, construction sand, and gravel were all placed into petri dishes such as that in Figure 8c, and measured as close to the sphere port as possible without risk of contaminating the inside of the sphere. Due to the uneven shapes and surfaces of the gravel, some gaps were observed between the sample and sphere (distances < 10 mm) as shown in Figure 8d. However, as with the grass, it was determined as preferable to measure the sample unaltered rather than change the structural composition so as to best mimic field conditions.

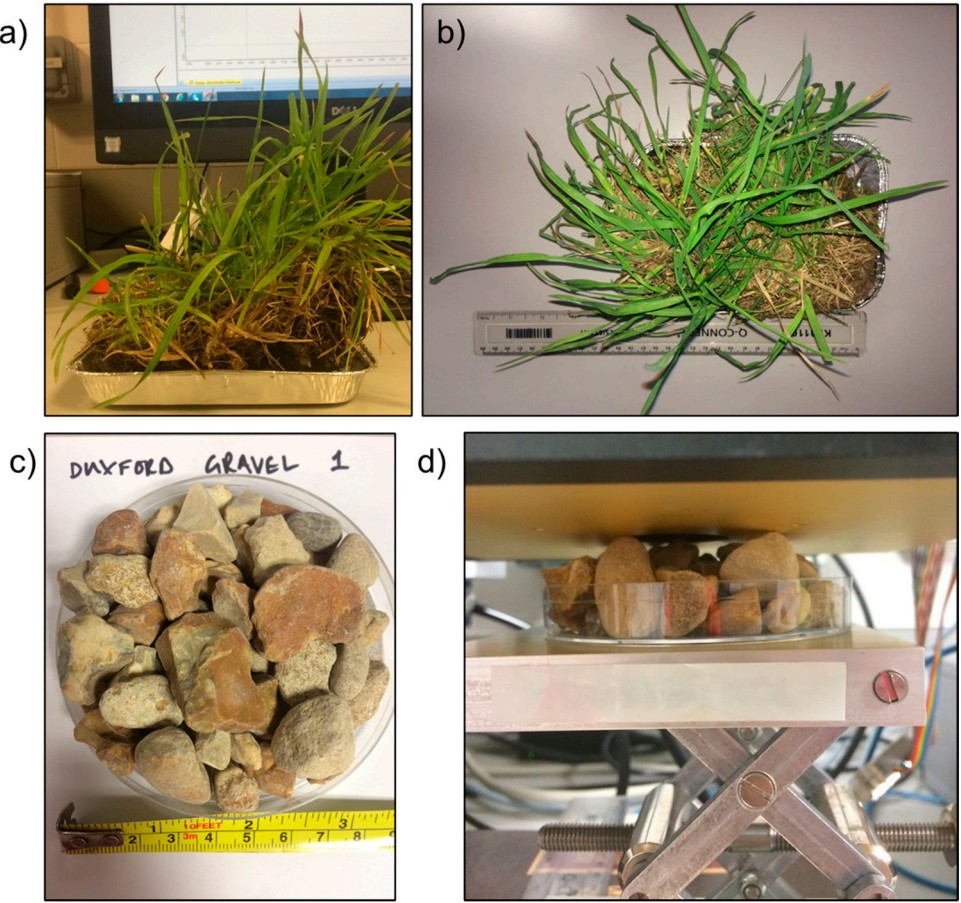

**Figure 8.** Laboratory preparation of (**a**) the grass sample from Alconbury from the side and (**b**) from above, (**c**) the gravel sample from Duxford from above, and (**d**) underneath the sample port of the external integrating sphere.

### 3.3. Emissivity Measurement Comparison

The surface spectral emissivities derived from measurements made by the two portable FTIR spectrometers (EM27 and D&P) along with those from the laboratory setup (Vertex 70) were compared to determine the absolute emissivity differences and the degree of agreement of the identified spectral features. The comparison was limited to the spectral range 8–13 µm, since this covers the wavelength range commonly employed in LST retrieval algorithms [62]. Measurements of the tarpaulins made using the EM27 in both Italy and the UK were compared to enable assessment of the performance of the same method in different environments.

Each FTIR-derived emissivity spectrum was then convolved with the Heitronics K15.85 radiometer spectral response function (Figure 4) to obtain broadband emissivity values comparable with those derived from the emissivity box.

### 3.4. Evaluation of Impact on LST

To understand the impact of any noted emissivity differences on LST estimation, a scenario was simulated for a near-surface Heitronics KT15.85 radiometer observing six samples measured in Grosseto and Duxford. Atmospheric transmissivity and path radiance effects were negligible due to the near-surface nature of the simulated observations, and sample-specific input values for land surface BT and sky BT were taken from in-situ LWIR radiometer measurements collected during the campaign (Table 5). Input emissivities ($\varepsilon$) used were the broadband emissivities derived for each of

the measurement methods used to assess the emissivity of that sample. LSTs corresponding to the radiometer were calculated for each sample input emissivity as in Guillevic et al. [3]:

$$LST = B^{-1}\left[\frac{1}{\varepsilon}\left(L_{\text{surf}} - (1 - \varepsilon)L^{\downarrow}_{\text{sky}}\right)\right]$$  (10)

where $B^{-1}(L)$ is the inverse Planck function describing the blackbody equivalent temperature ($T$, kelvin) of spectral radiance ($L$, W.m$^{-2}$.sr$^{-1}$.μm$^{-1}$), $L_{\text{surf}}$ the spectral radiance (W.m$^{-2}$.sr$^{-1}$.μm$^{-1}$) corresponding to the input surface viewing BT, and $L^{\downarrow}_{\text{sky}}$ the downwelling atmospheric LWIR spectral radiance (W.m$^{-2}$.sr$^{-1}$.μm$^{-1}$) corresponding to the sky viewed BT. Uncertainties were calculated and propagated as in Ghent et al. [63] and detailed in Appendix A.

**Table 5.** Input surface viewing and sky viewing brightness temperatures used to simulate land surface temperature with the measured emissivities.

| Sample | Location | Surface Viewing BT (K) | Sky Viewing BT (K) |
|---|---|---|---|
| White Tarpaulin | Grosseto, IT | 300 | 250 |
| Black Tarpaulin | Grosseto, IT | 330 | 250 |
| Grey Tarpaulin | Grosseto, IT | 330 | 250 |
| Grass | Duxford, UK | 300 | 240 |
| Gravel | Duxford, UK | 310 | 240 |
| Tarmac | Duxford, UK | 320 | 240 |

## 4. Results

### 4.1. Emissivity Measurement Inter-comparison

#### 4.1.1. Spectral Emissivities

Emissivity spectra (8–13 μm) for the Alconbury surface samples are shown in Figure 9, as measured in the field using the EM27 and in the laboratory using the Vertex 70. Figure 10 shows the spectral emissivities of the white, grey, and black tarpaulin as measured in the laboratory using the Vertex 70 and in Grosseto and Duxford using the EM27 and D&P spectrometers. Field- and laboratory-derived emissivity spectra of the other samples from Grosseto and Duxford are shown in Figure 11.

Alconbury

Results from Alconbury (Figure 9) enable a direct comparison of the laboratory (Vertex 70) and field-measured (EM27) spectral emissivities. The closest absolute agreement is found for the graybody samples (grass and water), with high and spectrally flat emissivities within 1% of each other between 8 and 12 μm. The laboratory and field measurements of polystyrene and card are generally within 2% across 8–12 μm, but differences of up to 0.04 are observed at points for the polystyrene (for example around 9.56 μm). For the sand sample, the laboratory and field measurements are within 2% between 9.8 μm and 12 μm but there are differences of 10–15% in the restrahlen bands over 8.0–9.5 μm, with the laboratory measured emissivity higher than the EM27 field measured emissivity. For the other samples, between 8 and 12 μm the field-derived emissivities tend generally to be slightly higher than the laboratory-derived values. Beyond 12 μm, there is increased noise in the field-measured (EM27) emissivity spectra, which could be due to increased atmospheric effects at these longer wavelengths (Figure 4).

Some non-physical spectral emissivities (>1) are observed in the laboratory measurements of the graybody samples and in the field measurements of the grass shown in Figure 9. Increased uncertainties are also observed for both field and laboratory measurements of the graybody samples compared to the other samples. Given that the surface temperatures of water and grass are sensitive

to even low winds [17], increased field uncertainties and noise for these sample measurements are probably due to varying sample temperatures during the measurement. The emissivities greater than unity found in the laboratory measured data appear also to be largely due to noise. Increased noise for these samples is expected due to the limitation of measuring samples of high spectral emissivity (low spectral reflectance) using a laboratory setup operating in directional hemispherical reflectance mode. An alternative explanation of the non-physical emissivities of the grass sample for both field and laboratory measurements could be canopy scattering, with increased emissivities due to the cavity effect [16].

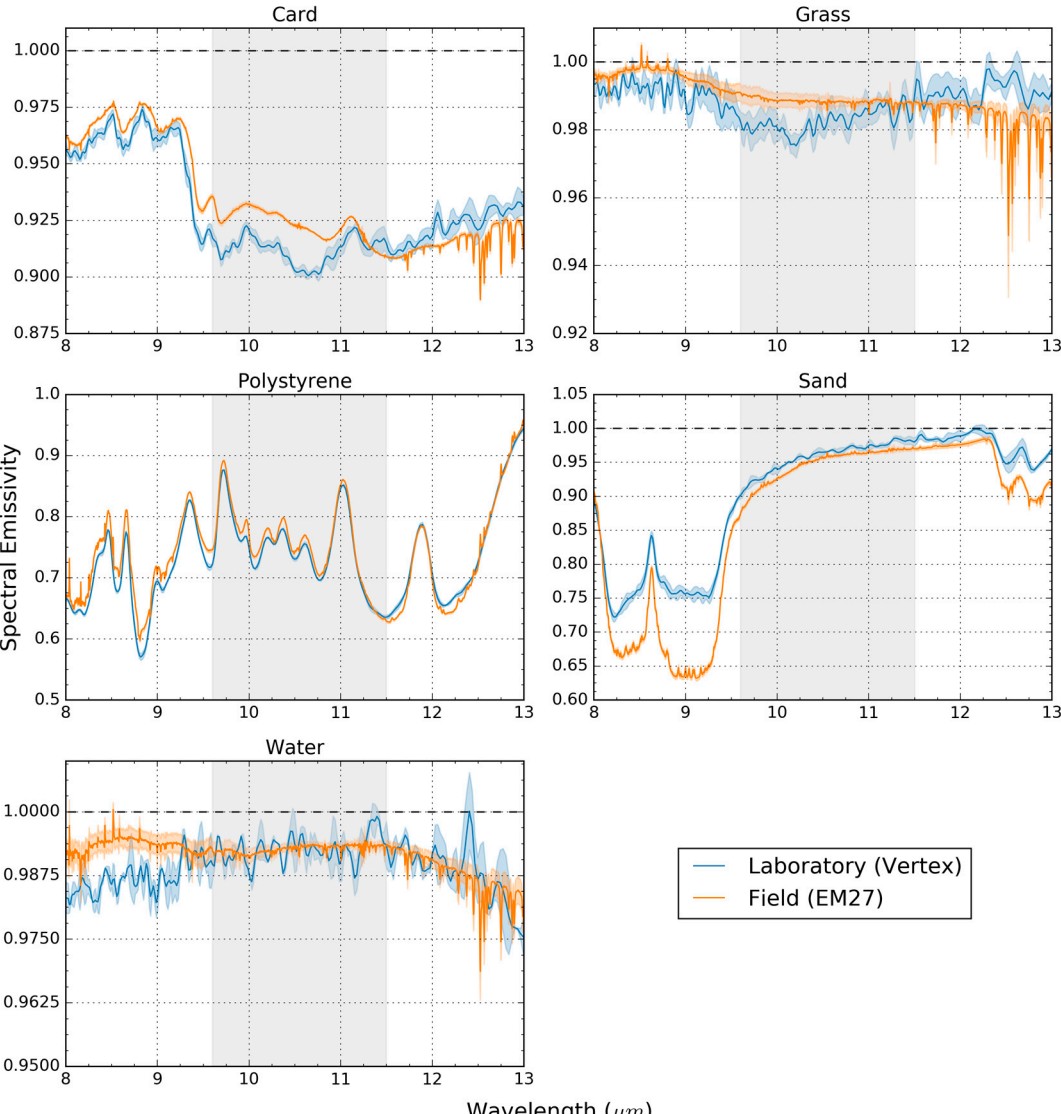

**Figure 9.** Laboratory-measured (Vertex) and field-measured (EM27) surface spectral emissivities of five different samples, as measured in Alconbury (UK) in May 2018. Values are the mean of all measurements, with the surrounding shaded area indicating the corresponding uncertainty as detailed in Section 3.1. The numbers of measurements made of each sample were listed in Table 4. Grey shaded area indicates the spectral range of the Heitronics KT15.85 IIP radiometer used for the emissivity box measurements.

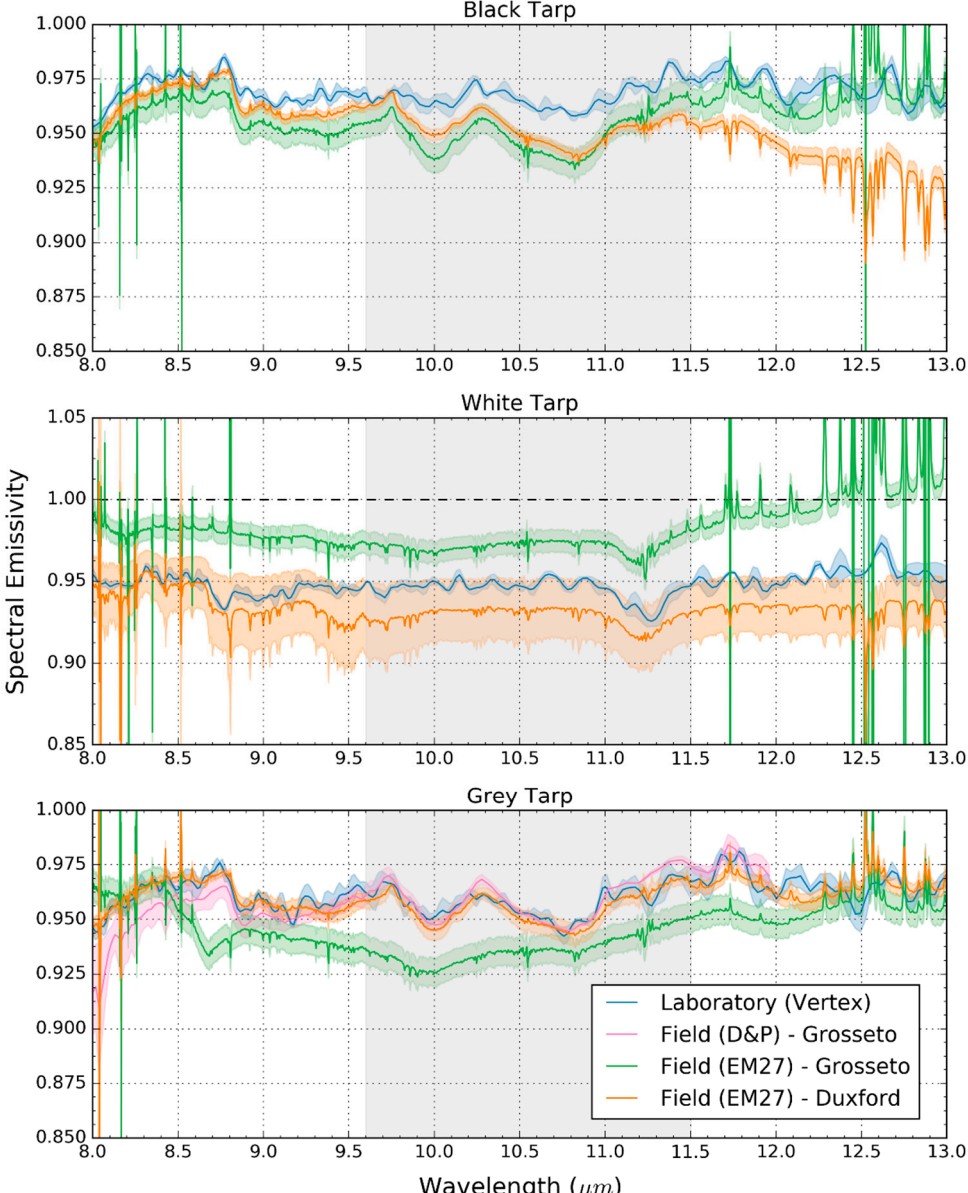

**Figure 10.** Spectral emissivities of (top to bottom) the black tarpaulin, white tarpaulin, and grey tarpaulin based on data collected in the laboratory (Vertex) and field using (i) the Bruker EM27 FTIR spectrometer (EM27) in both Grosseto, Italy and Duxford, UK and (ii) the Designs and Prototypes μFTIR spectrometer (D&P, grey tarpaulin only). Values are the mean of all measurements, with the surrounding shaded area indicating the corresponding uncertainty as detailed in Section 3.1. The numbers of measurements made of each sample were listed in Table 3. Grey shaded area indicates the spectral range of the Heitronics KT15.85 IIP radiometer used for the emissivity box measurements.

Considerable spectral variability is observed in the three non-graybody surfaces measured at Alconbury, with emissivities going down to about 0.6 (polystyrene, ~8.8 μm). The restrahlen bands (8–9.5 μm) and the Christiansen peak near 12.3 μm are clearly evident in both field and laboratory spectra of sand, although the minima in the restrahlen bands are weaker in the laboratory measurement. Despite absolute differences, the wavelengths at which specific spectral features are observed at correspond very well between the field (EM27) and laboratory (Vertex 70) measurements of the non-graybody samples, particularly for the polystyrene.

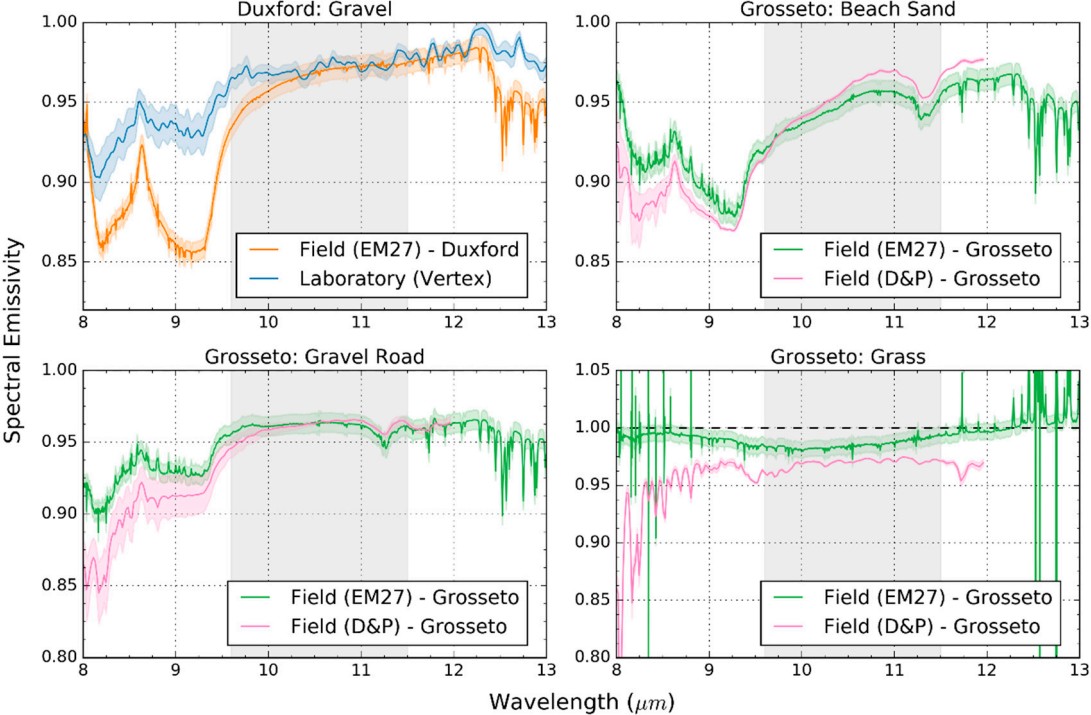

**Figure 11.** From top left clockwise, spectral emissivity measurements of (i) gravel from Duxford, (ii) beach sand in Grosseto, (iii) the sandy gravel drive in Grosseto, and (iv) short grass in Grosseto. Measurements were made using a Bruker Vertex 70 laboratory setup, a Designs and Prototypes μFTIR spectrometer (D&P) operated in the field, and a Bruker EM27 also operated in the field. Values are the mean of all measurements, with the surrounding shaded area indicating the corresponding uncertainty as detailed in Section 3.1. The numbers of measurements made of each sample were listed in Table 4. Grey shaded area indicates the spectral range of the Heitronics KT15.85 IIP radiometer used for the emissivity box measurements.

Grosseto and Duxford

Considering the spectral measurements of the samples from Grosseto and Duxford, the measurements of the tarpaulin made in the laboratory and those collected using the D&P μFTIR in Grosseto and the EM27 in the field at Duxford are all within 2% of each other between 8.0 and 12.0 μm (Figure 10). These differences are in line with the Alconbury measurements, and comparable with other studies that have compared emissivity measurement approaches [11,15]. However, agreement between the laboratory and field measurements of the gravel sample from Duxford is poor by comparison, with a difference of up to 8% observed between the EM27 (field) and Vertex (lab) measurements in the restrahlen bands between 8.0 and 9.5 μm (Figure 11). Furthermore, as with the sand measurement from Alconbury shown in Figure 9, while the restrahlen bands are clearly evident in the EM27 measurements of the gravel in Duxford (Gravel_Dux), these minima are weaker in the laboratory measurements.

The increase in noise in the derived spectral emissivity data beyond 12 μm for field measurements made using the EM27 in Alconbury (Figure 9) is again observed for all EM27 field measurements in Grosseto and Duxford (Figures 10 and 11 respectively). The EM27 measurements of the white tarpaulin and grass in Grosseto additionally show non-physical emissivities (>1) above 12 μm, which appeared systematic and not attributable solely to noise. Conversely, a decrease in spectral emissivity above 12 μm is observed in the EM27 measurements of the gravel in Grosseto, beach sand in Grosseto, and gravel in Duxford (Figure 11).

Spectral emissivity data of the same tarpaulins measured in the field in Duxford and in Grosseto using the same EM27 setup show larger differences than anticipated, in both spectral shape and

magnitude (Figure 10). The Duxford EM27 measurements were in better agreement with the laboratory- and D&P-measured spectra than the Grosseto EM27 measurements, particularly so for the grey tarpaulin where the EM27 measurements collected in Grosseto failed to identify certain spectral features. The EM27 measurements in Duxford by contrast were in close agreement (<1%) to those of the D&P (from Grosseto) and the laboratory measurements. Despite the Duxford measurements appearing to perform relatively better than those from Grosseto, high uncertainties are also observed in the Duxford spectral emissivity measurements of the white tarpaulin. The PVC coating of this particular target had slightly specular characteristics, which may have made the emissivity more variable between measurements as the EM27's retrieval method is intended for samples with Lambertian behaviour surfaces.

The sand, grass, and gravel samples from Grosseto shown in Figure 11 enable direct comparison of the data from the two portable FTIR spectrometers, with measurements collected almost simultaneously and under identical field conditions. The spectral emissivities of sand derived with the EM27 and the D&P instruments were within 1% of each other between 8.5 and 12.0 μm, and the gravel emissivities were within 2% of one another over the same range, with the increased differences for the gravel likely attributable to the increased variability within this material. There was also strong agreement seen between the spectral features for the beach sand and gravel road. These results promote confidence in the emissivity data derived from both FTIR instruments over the 8.5–12.0 μm range. While the measured emissivities of the grass from the two FTIR spectrometers were also within 2% of each, non-physical (>1) noisy emissivities are observed in the EM27 data of the grass sample in Grosseto, as was also observed in this instrument's measurement of the Alconbury grass sample (Figure 9).

Below 8.5 μm, spectral emissivities retrieved using the D&P μFTIR spectrometer seem to be consistently lower than those derived using the EM27, particularly for the grass sample (Grass_Gro) in Figure 11. This could indicate insufficient correction of atmospheric features in the post-processing of the D&P data, since this region has increased absorbance from atmospheric water vapour [17]. Outside this spectral region, EM27-derived emissivity spectra appeared consistently noisier than those from the D&P, as can be observed again in the grass measurements from Grosseto (Figure 11).

### 4.1.2. Broadband Emissivities

The derived broadband emissivities for all samples measured in Grosseto and Duxford with the EM27 and D&P systems are shown in Figure 11, alongside those derived using the two-lid emissivity box. Agreement between the FTIR-derived values and those of the emissivity box is excellent for some samples, such as the gravel road in Grosseto (Gravel_Gro) where the EM27 and D&P measurements were within 0.1% (0.001) of those of the emissivity box (Figure 12). However, for other samples the emissivity box provides broadband emissivities consistently lower than those of the FTIR systems, with differences of over 5% (0.05) for the grey tarpaulin for example (whereas for this sample the EM27 in Duxford, D&P in Grosseto and the laboratory Vertex 70 deliver broadband emissivities within 0.5% of each other).

Considering the different sample types, we observe that there were consistent discrepancies between the measurements of the two grass samples collected in Grosseto and Duxford using the EM27 and the emissivity box. For both, the emissivities of the vegetation as measured by the box method had a negative bias compared to the EM27 (Figure 12), with the EM27-derived values more in line with vegetation measurements reported elsewhere [39,51]. However, the measurement of the grass sample in Grosseto collected using the D&P was similar to the box-derived emissivity of that sample. Given this unclear performance and the non-physical emissivities observed in the EM27-derived emissivities of the grass samples from Alconbury (Grass_Alc, Figure 9) and Grosseto (Grass_Gro, Figure 11), further work is recommended to understand the performance of the EM27-based system over such targets.

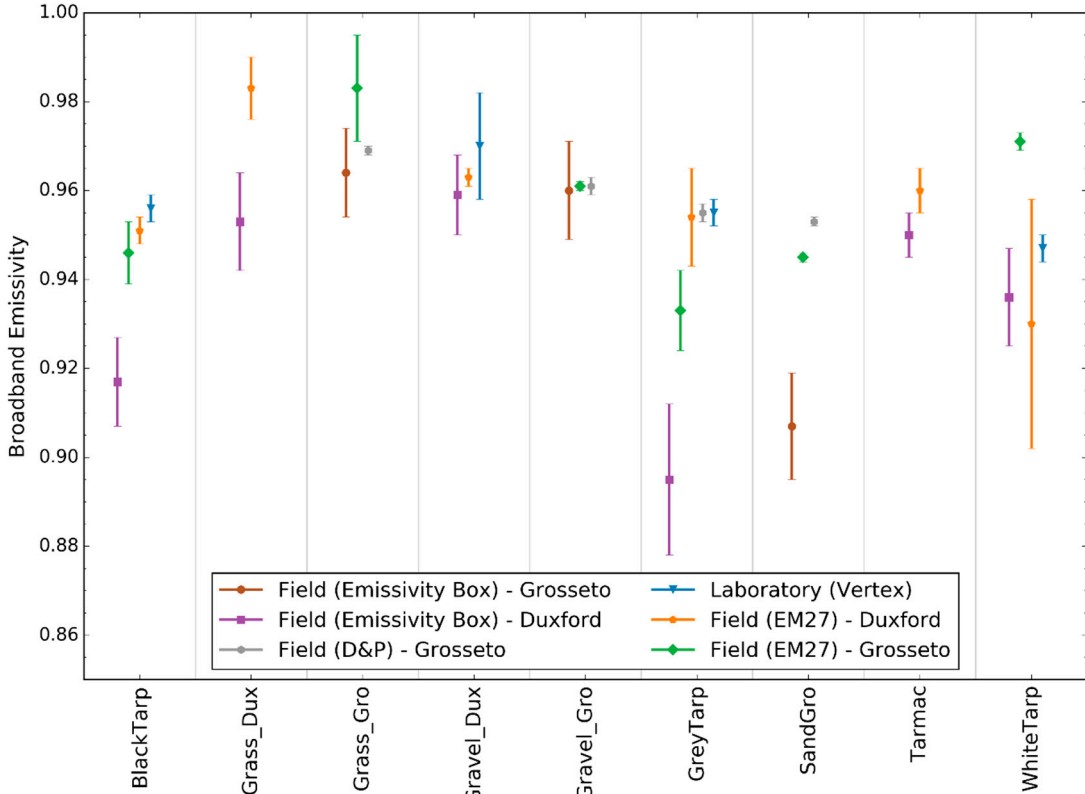

**Figure 12.** Broadband LWIR emissivities of target samples measured in Grosseto and Duxford by the EM27 and D&P FTIR systems, derived via the convolution of these surface spectral emissivity data with the spectral response function of the Heitronics KT15.85 radiometer shown in Figure 4. Matching broadband emissivity values derived using a two-lid emissivity box and that same Heitronics radiometer are shown alongside. Error bars show the uncertainties of each estimate.

Uncertainties were consistently around 0.010 for the box emissivity measurements, which is comparable with other studies making use of the emissivity box [15,37,52]. However, these were consistently higher than uncertainties from the other methods (with the exception of the white tarpaulin measurement collected using the EM27 in Duxford, which had a band-averaged uncertainty of 0.028). The increased uncertainty, and the lack of consistency found here in the relative emissivity values in comparison to the range of other sensors deployed, lead us to question the ability to reliably use the box method—and thus its suitability for calibration and validation studies.

### 4.2. Impact of Measurement Differences on LST Estimation

Table 6 shows the LST values and uncertainties calculated as detailed in Section 3.4 for each sample and input emissivity, with maximum and minimum derived LSTs highlighted in red and blue respectively. LSTs derived using emissivities from the two-lid emissivity box were the highest for all samples other than for the white tarpaulin, reflecting the consistent negatively biased emissivities delivered by this method relative to the others. In the case of the grey tarpaulin, the derived LSTs using the box-derived emissivities were 3.92 °C higher than those based on emissivities from the D&P μFTIR, EM27, or Vertex FTIR spectrometer as inputs. The magnitude of this bias again questions the reliability of the emissivity box approach.

**Table 6.** Calculated LSTs and LST uncertainties (°C) for the six samples, each calculated using the emissivities derived with the various field and laboratory emissivity measurement methods considered herein, with the median and interquartile range (IQR) for each sample.

| Sample | Calculated LST (°C) | | | | | | |
|--------|---------|------|----------------|----------------|------|--------|------|
| | TL-Box | D&P | EM27 (Grosseto) | EM27 (Duxford) | Lab | Median | IQR |
| WhiteTarp | 29.48 ± 0.38 | | 28.00 ± 0.36 | 29.74 ± 0.45 | 29.00 ± 0.37 | 29.24 | 0.80 |
| GreyTarp | 63.44 ± 0.39 | 59.52 ± 0.37 | 60.91 ± 0.39 | 59.59 ± 0.38 | 59.52 ± 0.37 | 59.59 | 1.39 |
| BlackTarp | 61.95 ± 0.38 | | 60.09 ± 0.38 | 59.77 ± 0.37 | 59.46 ± 0.37 | 59.93 | 0.86 |
| Gravel_Dux | 28.72 ± 0.37 | | | 28.52 ± 0.36 | 28.20 ± 0.37 | 28.53 | 0.26 |
| Grass_Dux | 39.23 ± 0.37 | | | 37.70 ± 0.36 | | 38.49 | 0.79 |
| Tarmac | 49.75 ± 0.37 | | | 49.15 ± 0.37 | | 49.45 | 0.30 |

Differences between LSTs calculated using the broadband emissivities derived from the FTIR-based methods deployed herein are smaller than those resulting from use of the box-derived emissivities. However, we still observe LSTs differing by up to 1 °C (white tarpaulin), which, given that the GCOS target accuracy and currently achievable requirements for LST as an ECV are 1 °C and 2–3 °C respectively [28], highlights the continuing importance of reducing uncertainties on emissivity retrieval.

## 5. Discussion

Comparison of the spectral emissivities found the majority of emissivities from field and laboratory spectrometers to be within 1–2% of each other for most of the spectral range 8.5–12.0 μm. These levels are broadly in line with other studies [11,12] and give confidence in the measurements and methods for selected samples. Outside of 8.5–12.0 μm, differences between emissivity measurements increased and increased noise levels were observed in the EM27 spectra. A potential cause for the reduced performance of the EM27 beyond 12 μm could be extrapolation error from calibration to the cold sky temperatures, which Korb et al. [31] observed to create spectral artefacts between 11 and 13 μm due to nonlinearity of the MCT detector responses over wide signal ranges. However, if this were the case, this decrease in emissivities beyond 12 μm would likely be observed in the EM27 measured spectra of all samples, which was not the case. Furthermore, no systematic distortion in the EM27 spectra was apparent at wavelengths below 12 μm as would likely be if this were the cause. While emissivities from 8.5 to 12.0 μm were satisfactory for most applications from field to satellite scale, with the majority of satellite thermal bands used for LST calculation located in this region [2], further investigation is required if field emissivities accurate outside of this range are required.

The differences between the laboratory (Vertex 70) and field (EM27) measurements in the restrahlen bands of the sand sample from Alconbury (Figure 9) and the gravel sample from Duxford (Figure 11) raise questions about the relative performances of these laboratory and field setups in this region for samples with strong restrahlen features. Further investigation in particular is recommended for the laboratory setup, since the EM27 measurements of the sand and gravel drive samples in Grosseto compared well with the D&P measurements of the same samples (Figure 11). In the case of the gravel sample from Duxford, a likely cause of this lack of agreement is the inhomogeneity of the sample (Figure 8c) together with the different field-of-regards between the two measurement techniques, with the diameter of the sample port in the laboratory setup half that of the field-of-regard for the EM27 in the setup deployed in Duxford (diameters of 25 mm and 50 mm respectively). A contributing factor to the higher emissivity in the laboratory measurement could also be that the gap between the gravel sample and sample port in the laboratory (discussed in Section 3.2.1) decreased the measured sample reflectance and increased the derived emissivity. Although neither of these interpretations are applicable to the sand measurements, these issues highlight the impact that the different scales and designs of laboratory and field instrumentation can have on the retrieved emissivity. This is particularly important when using emissivity from in situ measurements for calibration/validation

activities over targets such as gravel that are apparently homogeneous in satellite/airborne sensor pixels but heterogeneous at the sub-pixel level [64]. In this study, field measurements were found to be vital for such samples as they observe a larger area than laboratory measurements and can more easily cope with the target sample structure.

A key limitation of this study was that only one sample was measured by all four methods (GreyTarp, Figure 9). With this sample however, strong agreement was seen between the laboratory, D&P and EM27 (Duxford) while the broadband emissivity from the emissivity box measurement of the sample was considerably lower. The negative bias in emissivity box measurements of this sample and other samples compared to the other methods as shown in Figure 12 questioned the reliability of the emissivity box approach for calibration and validation activities, particularly since use of the emissivities from the box in this study were found to result in positive biases of up to 4 °C when used to simulate in situ LSTs from radiometer observations.

The differences between the EM27 measurements of the tarpaulins made in Grosseto and Duxford indicate field conditions have a strong impact on output emissivities. Interestingly, the EM27 measurements of the tarpaulin from Grosseto were found to be in poorer agreement with the laboratory and D&P measurements from Grosseto than the EM27 measurements of the tarpaulin from Duxford, despite being made at the same time and location as the D&P measurements. However, as shown in Figure 11, other measurements made with the EM27 and D&P in Grosseto were in closer agreement, with the D&P and EM27 measurements of gravel, grass, and sand made in Grosseto within 2% across 8.5–12 µm. More measurements with the D&P (e.g., in Duxford) would have assisted here in understanding the relative performance of these two field spectrometers. Increased noise in the EM27 measurements of the grass sample compared to the D&P measurements was likely an artefact of the different instrument spectral resolutions (EM27 at 0.5 cm$^{-1}$ and D&P at 4 cm$^{-1}$). However it could also indicate reduced sensitivity in the EM27 instrument compared to D&P. The latter interpretation was identified as the calculated sample temperature of the grass that was five degrees lower than the measured ambient air temperature (30.5 °C), thus reducing signal to noise [41]. This should be investigated further since if this is the case the performance of the EM27 will be limited for measurement of cold samples in hot environments due to the reduced signal to noise ratio for these samples unless modifications are made to improve the sensitivity of the instrument.

Two potential causes were identified for the poor agreement in the EM27 tarpaulin measurements from Grosseto compared to those from Duxford. Firstly, the calculated temperatures of the white and grey tarpaulin were within just 2 °C of the cold blackbody temperature used in the EM27's two temperature calibration, with Hook and Kahle [40] finding that absolute errors in field emissivity measurements increased where the sample temperature was close to the temperature of the calibration blackbody. Care should therefore be taken to ensure the blackbody temperatures are at least 5 °C either side of the estimated sample temperature, but the high ambient temperatures in Grosseto meant the power required to cool the blackbody to the necessary temperature was insufficient—an issue now resolved by the use of a more powerful inverter in the EM27 setup. This could also have been the cause of the high uncertainties in the EM27 measurement of the white tarpaulin from Duxford (Figure 10), since the calculated sample temperature was also only 0.5 °C above the cold blackbody temperature. A second potential cause of the differences in the tarpaulin measurements made in Grosseto and Duxford using the EM27 was identified through comparison of the measured downwelling radiances and humidity in Grosseto and Duxford (not shown). Increased downwelling radiances were observed throughout the Grosseto campaign compared to Duxford, which corresponded with increased humidity (Section 3.2). Furthermore, greater variability between consecutive measurements of the gold panel in Grosseto than in Duxford indicates downwelling radiances were changing more rapidly with time in Grosseto—providing increased scope for changes between the sequential measurements of the panel and the sample being collected. Environmental conditions therefore indicate reduced stability and increased humidity, both factors known to impact the accuracy of retrieved emissivities [17,31]. This interpretation was supported by the increased atmospheric emission lines between 8 and 9 µm

apparent in the EM27 spectra collected in Grosseto compared to Duxford (Figure 10). This may have impacted the EM27 measurements more than it did the D&P measurements due to (i) the higher spectral resolution and (ii) the spectral smoothness retrieval method used to derive emissivity from the EM27, which relies on minimizing atmospheric features across 8.12–8.60 µm and therefore is optimal for stable atmospheres. Consideration of the tarpaulin emissivity measurements derived from the two different locations (Grossetto and Duxford) therefore highlights that measurement accuracies and uncertainties were highly sensitive to environmental conditions, and that care should be taken to ensure the blackbody calibration temperatures properly bracket the sample temperature.

The measurement of vegetation was shown to prove challenging for all methods, with non-physical emissivities and high noise levels observed and differences between the broadband emissivities. In the case of the EM27, non-physical emissivities and high noise were attributed to increased variability in sample temperatures during measurement [57]. However, we also observed that the calculated sample temperature of the Alconbury grass (Grass_Alc, Figure 9) was just 1 K hotter than the cold blackbody temperature. As discussed above with the tarpaulin measurements, this could have led to increased errors. With respect to the laboratory measurements, while non-physical emissivities and high noise levels were observed in the measurements of the Alconbury grass (Figure 9), it is difficult to determine if there is a systematic problem with measurement of vegetation or if this was an exception as only one sample was measured in the laboratory. Measurements of additional vegetation samples in the laboratory would have enabled further analysis but no samples were collected in Grosseto and Duxford, as they would have deteriorated before measurement in the laboratory due to the gap between the collection date and measurement date. An increased number of scans would improve the signal-to-noise ratio and is therefore recommended for future measurement of low reflectance samples. However it should also be considered whether a setup operating in the DHR mode could be used to make measurements of vegetation, since vegetation tends to have non-isothermal properties (with different temperatures in different parts of the sample) but Kirchhoff's law theoretically requires samples to be isothermal [47]. Salisbury and D'Aria [48] avoided this by cutting vegetation samples and arranging them in a continuous monolayer on an adhesive tape substrate. However, this is also known to impact the emissivity by changing the structural composition and does not take into account any exposed soil components [65]. The non-isothermal properties of vegetation samples could also be the cause of the non-physical emissivities of the grass sample for the EM27-derived field measurements (which assumes a uniform sample temperature to calculate emissivity with the spectral smoothness method). This supports Ribeiro de Luz and Crowley's [47] argument for development of radiative transfer models that account for non-isothermal structures. Given that one of the major applications of LST from satellite and airborne sensors is monitoring evapotranspiration and crop health [66], further work on measurement of vegetation samples in both the laboratory and the field is therefore recommended. This is particularly important since the vegetation samples considered in both Duxford and Grosseto were limited to homogeneous short cropped grass, while in reality more complex samples containing exposed soil and more complex canopy structures are likely to also need assessing.

This study considered the impact these emissivity differences would have on LST algorithm validation activities through simulating in situ LSTs from field radiometers. However, in situ emissivity values from the laboratory or field instrumentation are also important for the development of LST and land surface emissivity (LSE) retrieval algorithms from satellite or airborne sensors, despite the development of new hyperspectral and multispectral thermal sensors and new physical retrieval algorithms (e.g., [26,67]) capable of simultaneous LST/LSE retrieval without the need for input emissivity estimates from land cover maps or other sources [62]. An example of such an application is in derivation of the coefficients for the Maximum–Minimum Difference (MMD) module in the TES algorithm [26] used to produce the operational Moderate Resolution Imaging Spectroradiometer (MODIS), ASTER, and ECOSTRESS LST/LSE products [53,68]. In this case, a negative bias in emissivity inputs would cause reduced maximum emissivities for the same min–max difference, thus shifting the regression curve, changing the coefficients in the MMD module and impacting the retrieved LSTs and LSEs. It is

crucial therefore for LST and LSE retrieval algorithm development and validation activities that work continues on improving and understanding uncertainties surrounding in situ emissivity measurement methods in the field and laboratory.

## 6. Summary and Conclusions

We conducted an inter-comparison of four different methods of LWIR surface emissivity retrieval, encompassing methods that derived full spectral emissivity data and broadband emissivities, and which operate in the field and in the laboratory. The methods considered are based on field measurements made with two portable FTIR spectrometers (a Bruker EM27 and a D&P μFTIR) operating in the emission mode, a laboratory FTIR spectrometer (Vertex 70) operating in directional hemispherical reflectance mode, and a two-lid emissivity box based on the design of Rubio et al. [33] also deployed in the field. Fourteen target samples were considered across four field sites covering both the UK and Italy, and these include man-made materials such as tarpaulins and natural materials such as sand, grass, and water.

The majority of the derived spectral emissivities were within 1–2% of each other between the major part of the LWIR atmospheric window (8.5–12.0 μm), with identification of spectral features also in agreement between the different field and laboratory approaches. This degree of agreement is consistent with that found by other studies comparing field and laboratory methods of spectral emissivity determination. Differences of up to 15% were observed between the laboratory and field measurements for samples with strong restrahlen features, suggesting a need for further investigation into the laboratory setup's performance when measuring samples with these features. Consideration of the gravel sample from Duxford suggests that field instrumentation can be more suitable than laboratory directional hemispherical reflectance setups for non-homogeneous samples and samples with complex structures. Beyond 12 μm, significant noise and an unexplained drop off in spectral emissivity was observed in certain of the EM27 retrieved emissivities. As a result, we recommend use of EM27 emissivity spectra should be limited to within the 8.0–12.0 μm region. Similarly, although fewer measurements were made using the D&P, increased noise and a decrease in emissivity below 8.5 μm indicates that the D&P-system may deliver emissivities not fully to be trusted below this wavelength, at least in the configuration used herein.

Differences between field measurements made of the same samples using the EM27 but in different locations under different environmental conditions identified some issues. In particular the power supply was inadequate to cool the internal blackbody to the ideal temperature when ambient conditions were particularly warm, leading to the cold blackbody temperature being probably too similar to the target sample temperature to give well calibrated data. This has now been resolved through installation of a higher power inverter. Some increased noise was also evident in certain EM27 measurements, and we recommend that for comparatively cool samples such as vegetation data collection should be done at times to maximise thermal contrast with the surroundings. The time taken to collect each spectral measurement should also be minimised under conditions of potentially changing atmospheric humidity, for example by reducing the number of scans or lowering the measurement spectral resolution (Salisbury [17] advise that 8 cm$^{-1}$ is generally adequate for spectral emissivity determination).

Measurement of vegetation samples was found to be challenging for all methods due to reduced signal-to-noise, canopy scattering, varying sample temperature during the measurement and non-isothermal properties. Using the measured emissivities to simulate near-surface LST observations of grass found differences of 1.5 °C depending on which method of emissivity determination was used. Given that a major application of LSTs is for agriculture and use in evapotranspiration models [6], accurate measurement of the emissivity of vegetation at the field and laboratory scale is crucial, so further work towards understanding the uncertainties at both the field and laboratory scale is recommended.

We derived broadband emissivities from the spectral emissivity measurements and compared these with those calculated using the two-lid emissivity box method. We found a lack of consistency

in the emissivity values measured with the box and increased uncertainties compared to the other methods. This indicates that its performance was inferior to that of the FTIR-based approaches, albeit it is based on far cheaper and more available technology.

**Author Contributions:** Conceptualization, M.F.L., T.P.F.D. and M.W.; Data curation, M.F.L., T.P.F.D., M.W., M.J.G., M.C.d.J. and W.R.J.; Formal analysis, M.F.L.; Funding acquisition, T.P.F.D. and M.W.; Investigation, M.F.L., T.P.F.D., M.W., M.J.G., M.C.d.J. and W.R.J.; Methodology, M.F.L., T.P.F.D. and M.W.; Project administration, M.F.L., T.P.F.D. and M.W.; Resources, M.F.L., T.P.F.D., M.W., J.J., W.R.J., S.J.H. and G.R.; Software, M.F.L.; Supervision, M.F.L., T.P.F.D. and M.W.; Validation, M.F.L.; Visualization, M.F.L. and T.P.F.D.; Writing—original draft, M.F.L., T.P.F.D. and M.W.; Writing—review and editing, M.F.L., T.P.F.D., M.W., M.J.G., S.J.H. and G.R. All authors have read and agreed to the published version of the manuscript.

**Funding:** Aspects of this work were part of the joint NASA ESA Temperature Sensing Experiment (NET-Sense) conducted under a programme of, and funded by, the European Space Agency (Contract Number 4000131017/20/NL/FF/ab) and the National Aeronautics and Space Administration (NASA), with part of the research described in this paper carried out in part at the Jet Propulsion Laboratory, California Institute of Technology, under contracts with NASA. In addition, aspects were funded through PRISE (Pest Risk Information Service), a project funded by the UK Space Agency as part of the Global Challenge Research Fund. Support for this research also came partly from NERC National Capability funding to the National Centre for Earth Observation (NE/Ro16518/1).

**Acknowledgments:** We thank Hannah Nyugen, Bruce Main and Francis O'Shea from King's College London for their assistance in development and testing of the emissivity box on multiple field campaigns. The views in this publication can in no way be taken to reflect the official opinion of the European Space Agency or any other funding body.

**Conflicts of Interest:** The authors declare no conflict of interest.

## Appendix A

This appendix presents the calculation of uncertainties associated with the evaluation of the impact on LST presented in Section 3.4. The error sources on LST were identified as that of the surface radiation ($L_\uparrow$), downwelling radiation ($L_\downarrow$), and emissivity ($\varepsilon$). All terms are wavelength ($\lambda$) dependent but the wavelength terms were omitted for clarity.

To calculate the uncertainty on the derived LST observations, the equivalent uncertainties in radiance units $\left(U_{L\uparrow/\downarrow}\right)$ for both surface and sky viewing radiometer observations were first determined from the manufacturer stated uncertainty of the radiometer in temperature units ($U_{T\uparrow/\downarrow}$) through the differential of the Planck function with respect to temperature ($T$) such that:

$$U_L = \left|\frac{\partial B}{\partial T}\right|U_T = \frac{c_1 c_2 e^{\frac{c_2}{\lambda T}}}{\lambda^6 T^2 \left(e^{\frac{c_2}{\lambda T}} - 1\right)^2} \, U_T \tag{A1}$$

where $c_1$ and $c_2$ are constants such that $c_1 = 2hc^2$ and $c_2 = \frac{hc}{k}$ (with $h$, $c$ and $k$ as defined in Section 3.1.2).

The uncertainty of the land surface radiance ($U_{L_{surf}}$) was then calculated using Equation (8) in Ghent et al. [63] such that:

$$U_{L_{surf}} = L_{surf} \sqrt{\frac{U_{L\uparrow}^2 + \left((1-\varepsilon)L_\downarrow \sqrt{\frac{U_\varepsilon^2}{(1-\varepsilon)^2} + \frac{U_{L\downarrow}^2}{L_\downarrow^2}}\right)^2}{\left(L_\uparrow - L_\downarrow(1-\varepsilon)\right)^2 + \frac{U_\varepsilon^2}{\varepsilon^2}}} \tag{A2}$$

where $U_\varepsilon$ is the uncertainty on the emissivity observation. Using the uncertainty of the surface radiance, we then calculated the absolute uncertainty of a given LST observation ($U_{LST}$) using Equation (9) in Ghent et al. [63]:

$$U_{LST} = C_2 \left(\frac{c_1 \left(\frac{U_{L_{surf}}}{\lambda^5 L_{surf}^2}\right)}{\left(\frac{c_1}{L_{surf}\lambda^5} + 1\right) \lambda \left(\ln \frac{c_1}{L_{surf}\lambda^5} + 1\right)^2}\right) \tag{A3}$$

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
