# Peer review of "Inter-Comparison of Field- and Laboratory-Derived Surface Emissivities of Natural and Manmade Materials in Support of Land Surface Temperature (LST) Remote Sensing"

_remotesensing, doi:10.3390/rs12244127_

Round 1

Reviewer 1 Report

The submitted paper presents a comparison between different methods used to derive emissivity both in-situ and in the laboratory. To do so, 14 natural and manmade samples are measured using FTIR spectrometers, providing spectral emissivity, and an emissivity box, providing broadband emissivity. The differences observed between FTIR laboratory and field spectral emissivity measurements as well as between the broadband emissivity derived from FTIR measurements and the box method are analyzed, focusing on the 8-12 microns region. Finally the impacted on LST estimation is presented.

The issue addressed in this paper is very important knowing the impact of emissivity on LST estimation. It is crucial to study the performance of emissivity measurement methods to provide references and good practice advices. The application a large variety of material is interesting. This paper is well written and pleasant to read even if some results could be a bit more detailed. It is a bit late now but more measurements with the D&P would have helped for results interpretation. Please find below some specific comments about the article.

Section 4.1.1.1 Alconbury

There is no comment concerning the first graph of Figure 9 (“card”) while it seems that there is an emissivity difference of at least 3% at some wavelengths (e.g. 10.5 microns).

Is it really as mentioned in l.487 “Between 8 and 12 μm there is good agreement, with 486 differences of less than 2% for all samples other than sand”?

You may add a sentence about it.

Section 4.1.1.2 Grosseto and Duxford

According to the last graph of Figure 10, there is a better agreement between D&P in Grosseto and EM27 in Duxford. It is mentioned l.564. However, no explanation is provided why D&P has better agreement than EM27 both measured in Grosseto, with the EM27 measurement from Duwford. Maybe D&P would have had good agreement in all cases. More measurements with the D&P would have helped to conclude here as well as in section 4.2 - Impact of Measurement Differences on LST Estimation.

Minor comments

  • 141: “The conclusion…” instead of “These conclusion…”
  • 193: The last sentence is a bit confusing. As if comparison with laboratory measurements was not so important but it is in the title of the paper.
  • 256: Remove “and b)” at the end of the legend
  • 264: “Consecutive spectral measurements at were then made of the sample…”, not clear, please rephrase.
  • 321: It is the second Figure 2. From here, there is a shit in figure number
  • 333: “Laboratory calibration tests confirmed the 332 radiometer to have an absolute accuracy of ± 0.5 ° C, plus 0.7% of the difference between the 333 temperature of the target and that of the radiometer housing.”, not clear, please rephrase.
  • 337: Figure 3 is not really necessary but it is up to you
  • 359: Formatting error in Equation 7
  • 405: “construction sand from above”, what “from above” means here?
  • 412: you could add a column in the table to separate field and lab instruments, it would be easier to spot which samples were measured only in the field of both in the field and in the lab
  • 730: “We found a lack of consistency 729 in the emissivity values measured with the box, with together with the increased uncertainties found 730 with this method indicates that its performance is inferior to”, not clear, please rephrase.

Reviewer 2 Report

See attachment.

Reviewer 3 Report

GENERAL COMMENTS:

The study measured the emissivity of 14 artificial and natural samples in the field and in the laboratory to understand emissivity measurements under different conditions. Is the purpose of this comparison to identify any problems encountered in the measurement process or to suggest the most suitable emissivity measurement method? In my opinion, this study is valuable when it is linked to the uncertainty of LST measurements using satellite data. However, this study does not go into this in depth. Further discussion regarding satellite LST is needed.

It appears that this paper has not been sufficiently reviewed prior to submission. There are too many mistakes in writing manuscript. It does not follow the general paper writing method. You should refer to many other papers and follow the writing method. Therefore, in its present form, it is not suitable for publication in Remote Sensing. I recommend for revise and resubmit.

SPECIFICS COMMENTS:

L26 Fourier Transform InfraRed (FTIR) spectrometers

L28 each other? In situ and laboratory? Or FTIR Spectrometer and emissivity box?

L29-31 Add an explanation as to why the difference between laboratory and field measurements highlights the importance of field methods.

L38 Is (ε) needed in the introduction?

L85-87 Are there other studies that show that laboratory uncertainties are greater than those associated with field measurements or highlight the importance of field measurements? There must be more examples to use the word "typically".

L155 “as discussed in Section 1”, unnecessary mention

L166 Horton et al.’s

L168 Horton et al.

L182-183 “Those that have been conducted considered relatively few samples-” Consider rephrasing for clarity.

L187-193 These sentences should be moved to the introduction.

L195 varying structures? What does it mean?

L195-196 Why did the field campaigns take place in the UK and Italy? Is it related to the study purpose?

L215 The data coming from the laboratory system-

L219 on the system deployed in the current study and shown in Figure 1

→ on the same laboratory system

L224-225 The reflected spectral radiance (??(?)) is then measured and compared to a subsequent measurement of the reflected radiance from the internal wall of the integrating sphere (??(?)),

L226 Here ??(?) is an open → (no space) here ??(?) is an open

L231-233 Consider rephrasing for clarity.

L237 The first was the aforementioned Designs & Prototypes μFTIR spectrometer → The first was the D&P μFTIR spectrometer. The D&P abbreviation has already been used.

L256 b) ? Please write the caption clearly.

L250-251, 272 D&P μFTIR spectrometer

L281 where

L283 at temperature (Tpanel) calculated

L284 (εpanel)was → (εpanel) was

L285 the surface spectral emissivity (?(?)) of the sample

L288 where ???(??, ?) is the blackbody spectral radiance at temperature (??).

L293-294 multiple assumed realistic sample temperatures?

L296 8.12 – 8.60μm → 8.12 – 8.60 μm

L293-296 Consider dividing the sentence for clarity and understanding

L299 manufacturers figures?

L321 Change all figure numbers from figure 2

L328 described in [58]. Expressions like this are used too often (e.g., L241, 327, 343, 351…). See how to cite references in other papers.

L329 9.6μm to 11.5 μm → 9.6 – 11.5 μm

L346 where

L359 Eqn. → Eq.

L363 The emissivity → (space) The emissivity

L376 King’s College London

L412 Table 4 shows latitude and longitude in some samples only. What is the reason? Are latitude and longitude information important for this study?

L460 brightness temperature (BT) → BT, BT abbreviation has already been used.

L461 during the campaign, shown in Table 5. → during the campaign (Table 5).

L465 Where → (no space) where

L465-466 the blackbody equivalent temperature (T (Kelvin)) of spectral radiance (L (W.m−2.sr−1.μm−1)),

L469-470 Uncertainties were calculated and propagated as in [61]. This is not a correct citation. The equation used should be presented.

L499-504 The sentence is too long. Seperate the sentence for clarity.

L502 Remove “and thus”

L547 Korb et al.

L691-692 D&P μFTIR

L701-703 Does it just mean that field instrumentation measures emissivity more accurately than laboratory instrumentations? Give the difference in accuracy numerically. "the continuing importance of field-based methods"? Are field-based methods only important for certain materials such as gravel with a large difference? For common materials, the difference is 1-2%. For other common materials, isn't field-based method important?

Reviewer 4 Report

This manuscript provides an important contribution to the field of remote sensing of land surface temperature, as several of the retrieval algorithms currently in use rely on emissivity information from libraries of spectral measurements, whose uncertainty is not well known. The manuscript consists of an intercomparison of emissivity measurements using 4 different methods, both in field conditions and in controlled laboratory conditions. The methods are adequately described and their caveats are thoroughly discussed.

My only very minor request is regarding Lines 663-668 – I was a bit lost there, does this paragraph refer to any result shown in a figure/table? If not, maybe include a “not shown” note, so others don’t get lost too.

Apart from that, I think this manuscript is fully suitable for publication in Remote Sensing.